# Exploring through Random Curiosity with General Value Functions

**Aditya Ramesh**[1]   **Louis Kirsch**[1]   **Sjoerd van Steenkiste**[2*]   **Jürgen Schmidhuber**[1,3,4]

[1]The Swiss AI Lab (IDSIA), University of Lugano (USI) & SUPSI
[2]Google Research
[3]AI Initiative, King Abdullah University of Science and Technology (KAUST)
[4]NNAISENSE
{aditya, louis, juergen}@idsia.ch, svansteenkiste@google.com

## Abstract

Efficient exploration in reinforcement learning is a challenging problem commonly addressed through *intrinsic* rewards. Recent prominent approaches are based on *state novelty* or variants of *artificial curiosity*. However, directly applying them to partially observable environments can be ineffective and lead to premature dissipation of intrinsic rewards. Here we propose random curiosity with general value functions (RC-GVF), a novel intrinsic reward function that draws upon connections between these distinct approaches. Instead of using only the current observation's novelty or a curiosity bonus for failing to predict precise environment dynamics, RC-GVF derives intrinsic rewards through predicting temporally extended general value functions. We demonstrate that this improves exploration in a hard-exploration diabolical lock problem. Furthermore, RC-GVF significantly outperforms previous methods in the absence of ground-truth episodic counts in the partially observable MiniGrid environments. Panoramic observations on Mini-Grid further boost RC-GVF's performance such that it is competitive to baselines exploiting privileged information in form of episodic counts.

## 1   Introduction

How should an agent efficiently explore environments with high-dimensional state spaces and sparse rewards [70, 2]? An *extrinsic* reward-maximising agent will make little progress until it stumbles upon a rewarding sequence of actions. In the literature, this issue is commonly addressed by providing additional *intrinsic* reward to guide the agent's exploratory behaviour [49, 12, 43].

One class of prominent approaches rely on *state novelty*, where an intrinsic reward in the form of a 'novelty bonus' is awarded based on how often a state has been visited [65, 6]. More recent works have extended these approaches to high dimensional state spaces where tabular counts are inapplicable [7, 41, 10]. Another class of approaches is based on *artificial curiosity*, where agents are rewarded in proportion to the prediction errors or information gains of a predictive world model [52, 60]. Curiosity-based techniques have also been scaled up to handle larger state spaces [22, 43].

Expecting an agent to have access to the complete environment state is unrealistic. In such *partially observable* settings [3, 26], observations may look alike in different states, and the benefit of approximate state novelty bonuses as intrinsic rewards is unclear. Consider the environment in Figure 1 where an agent is required to traverse a series of blue and white tiles, yet it is only able to observe the current tile it is standing on. The colours of the tiles alternate in a regular fashion until the end of the corridor, where there is a surprising sequence of consecutive blue tiles. Directly

---

*Part of this work was done while the author was a Postdoctoral researcher at IDSIA.

36th Conference on Neural Information Processing Systems (NeurIPS 2022).

applying state novelty approaches may not be particularly meaningful in this case, as can be seen when using random network distillation (RND; [10]). In RND, the agent receives a novelty bonus based on the error of its predictions about the output of a fixed randomly initialised neural network applied to the current observation. As shown in the top panel of Figure 1, RND is unable to ascribe novelty when the state pattern changes later in the sequence and the intrinsic reward has vanished. Similarly, intrinsic rewards may also vanish prematurely for certain curiosity-based approaches when single-step dynamics are simple [42], while longer horizon predictions (as needed in the partially observable case) are susceptible to rapidly compounding errors during sequential rollouts [69].

In this paper, we explore a connection between state novelty and artificial curiosity to address these limitations. We propose *random curiosity with general value functions (RC-GVF)* as a novel approach to generating intrinsic rewards. It takes inspiration from exploration through temporally extended experiments that summarise long observation sequences [53]. Our approach uses random general value functions (GVFs) [67] to pose questions about the cumulative future value of observation-dependent features. Specifically, we train an ensemble of predictors to minimise the temporal difference (TD) errors of the general value functions and derive an intrinsic reward based on the TD-errors and disagreements in the long-term predictions. The effective horizon of the prediction task can be controlled via the discount factor, where a zero discount can be related to state novelty with random network distillation (RND) [10]. We hypothesise that predicting information from an temporally extended horizon

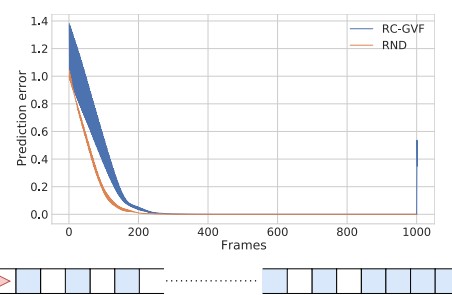

Figure 1: The alternating tile corridor environment. The agent (red triangle) observes the colour of the tile that it currently occupies (white or blue). The extrinsic reward is always zero. At each time-step (frame), the agent moves one tile forward, until it reaches the last tile. Unlike our approach (RC-GVF), RND does not generate an intrinsic reward when encountering the surprising last few tiles (step 1000).

improves exploration in POMDPs and guards against premature vanishing of intrinsic rewards due to the increase in difficulty of the auxiliary task. Indeed, in our toy experiment of Figure 1 it can be seen how RC-GVF manages to generate a spike of intrinsic reward at the end of the sequence.

We evaluate our approach on sparse-reward and partially observable RL problems. Our results on a hard exploration diabolical lock problem [36] and the Minigrid suite of environments [13] reveal the benefits of considering extended horizon predictions in comparison to approaches that rely on immediate partial observations to generate exploration bonuses. Furthermore, existing baselines in Minigrid are heavily reliant on privileged information about episodic state-visitation counts from the simulator, which is commonly used to scale the exploration bonus. We demonstrate that RC-GVF succeeds in many environments *without* the use of such episodic counts.

## 2 Preliminaries

**Reinforcement learning in POMDPs** We consider the scenario where an agent interacts with a Partially Observable Markov Decision Process (POMDP) with time steps $t \in \mathbb{N}$, with environment states $S_t \in \mathcal{S}$, observations $O_t \in \mathcal{O}$, actions $A_t \in \mathcal{A}$, extrinsic rewards $R_e : \mathcal{S} \to [0, 1]$, and policies that map histories of different lengths to distributions over actions $\pi : \mathcal{H} \to \Delta(\mathcal{A})$ where $H_t = O_{1:t} \in \mathcal{H}$ [26]. The agent only observes $O_t$ at time step $t$, which may not fully describe the MDP state, and thus it becomes necessary to condition the agent on the history $H_t$.

The learning objective is to find the optimal policy $\pi^*$ that maximizes the expected discounted return

$$J(\pi) = \mathbb{E}_\pi \left[ \sum_{k=0}^{\infty} \gamma^k R_e(S_k) \right], \tag{1}$$

where $\gamma \in [0, 1)$ is the discount factor and the expectation is over stochastic components in the environment and the policy.

**Random Network Distillation**  In random network distillation (RND) [10] the agent receives a state novelty reward proportional to the error of making predictions $\hat{z}(o_t)$ where the targets are generated by a randomly initialized neural network $Z_\phi : \mathcal{O} \to \mathbb{R}^d$. The RND intrinsic reward for an observation is given by

$$R_i(o_t) = \| Z_\phi(o_t) - \hat{z}(o_t) \|_2. \tag{2}$$

RND-like error signals can be a simple way to obtain effective uncertainty estimates about targets at a given input, which could subsequently be used for exploration in RL or detecting out of distribution samples [14]. Beyond its simplicity, an appealing aspect of using RND is that it allows us to implicitly incorporate prior beliefs about useful rewards through the neural network architecture [38].

**General Value Functions**  A general value function (GVF) [67] is defined by a policy $\pi$, a cumulant or pseudo-reward function $Z : \mathcal{O} \to \mathbb{R}$, and a discount factor $\gamma_z \in [0, 1)$. It can be expressed as

$$v_{\pi,z}(o) = \mathbb{E}_\pi \left[ \sum_{k=0}^{\infty} \gamma_z^k Z(O_{t+k}) | O_t = o \right]. \tag{3}$$

General value functions extend the concept of predicting expected cumulative values to arbitrary signals beyond the reward. They can be viewed as 'answering' questions about cumulative quantities of interest under a particular policy $\pi$ and discount factor $\gamma_z$. Predictions from GVFs have previously been used as features for state representation [47] or to specify auxiliary tasks that assist in shaping representations for the main task [24, 73].

## 3  Random Curiosity with General Value Functions

In the spirit of artificial curiosity [53], we are interested in predicting long-term outcomes in the environment under a particular policy as a means for exploration. Our approach, *RC-GVF*, rewards an agent for taking actions that lead to higher uncertainty about the future. However, instead of predicting the entire sequence of future states, we consider a set of random numeric questions about the environment and capture uncertainty about their outcome using general value functions (GVFs).

### 3.1  General value functions to predict the future

The central piece of RC-GVF is the prediction of temporally extended outcomes in the environment. This is different from only predicting a feature of the current state (as in state novelty exploration [65]) or dynamics models that predict the entire (sequence of) next observation(s) (as in early versions of artificial curiosity [52]). At time step $t$, the current observation $o_t$ is mapped to a collection of pseudo-rewards $z_{t+1} \in \mathbb{R}^d$ which together with the policy and discount factor define a question that can be asked about the future: "what is the expected discounted cumulative sum of these pseudo-rewards under a given policy?". The answer to this question (i.e. the prediction target) is based on outcomes in the environment, as in Equation 3. In this paper, we investigate whether a set of random pseudo rewards are sufficient to drive exploration in this way.

Previously it was found that random features extracted by a neural network are often sufficient to capture useful properties of the input [45]. Similarly, neural network architectures can be used to express prior knowledge about useful features [71, 14]. To this end, we generate pseudo-rewards from a fixed and randomly initialised neural network $Z_\phi : \mathcal{O} \to \mathbb{R}^d$ with parameters $\phi$ that maps observations to pseudo-rewards. This choice also bypasses the difficult problem of discovering meaningful general value functions [73].

**Training the GVF predictors**  Our exploration mechanism is to reward the agent for taking actions that generate previously unknown outcomes. To facilitate this, we train a separate (recurrent) neural network–which we call the *predictor*–to predict these values. Concretely, a predictor $\hat{v}_{\pi,z} : \mathcal{H} \to \mathbb{R}^d$ maps histories (of observations, actions, and pseudo rewards) to pseudo-values.

The predictor is trained on-policy, implying that the GVFs are evaluated under the current policy. One motivating factor for this choice is that it couples the prediction task to the current policy, thus creating an incentive to vary behaviour for additional exploration [18].

We use the (truncated) $\lambda$-return as the target for the predictor, which can be recursively expressed as

$$G_t^z(\lambda_z) = Z_{t+1} + \gamma_z(1 - \lambda_z)\hat{v}_{\pi,z}(H_{t+1}) + \gamma_z\lambda_z G_{t+1}^z(\lambda_z). \tag{4}$$

Here, $\lambda_z \in [0, 1]$ is the usual parameter that allows balancing the bias-variance trade off by interpolating between TD(0) and Monte Carlo estimates of the pseudo-return [64]. The predictor is trained to minimise the mean squared TD-error with the $\lambda$-return target. For convenience in notation, we will denote $G_t^z(\lambda_z)$ as $G_t^z$, despite its dependence on $\lambda_z$.

## 3.2 Disagreement and prediction error as intrinsic reward

To generate an intrinsic reward, a straightforward choice could be to consider the error between the temporal difference target (from Equation 4) and the predictor's output at the current observation:

$$L_{\text{TD}}(h_t) = [G_t^z - \hat{v}_{\pi,z}(h_t)]^2. \tag{5}$$

However, the presence of *aleatoric* uncertainty is a problem that arises as a consequence of extending the horizon of the predictive task. From an exploration perspective, the agent should focus on the reducible *epistemic* uncertainty, and not the irreducible aleatoric uncertainty [15, 23]. Directly minimising Equation 5 with the predictor would ignore the inherent variance in the TD-target (due to stochasticity in the policy and environment), and thus using the prediction error of the GVF target as an intrinsic reward does not distinguish between aleatoric and epistemic uncertainty.

Another way to recognize this is by decomposing the expected prediction error as described by Jain et al. [25]. The expected loss of a predictor $\hat{f}(\cdot)$ at an input $o$, $\mathbb{E}[L(t, \hat{f}(o))]$, can be split into epistemic ($E$) and aleatoric ($A$) components: $\int L(t, \hat{f}(o))dP(t|o) = E(\hat{f}, o) + A(o)$, where $P(t|o)$ is the conditional distribution over targets for input $o$ and $L(t, \cdot)$ is a chosen loss function. The aleatoric uncertainty at input $o$, $A(o)$, is defined as the expected prediction error of a Bayes optimal predictor function. $E(\hat{f}, o)$ denotes the epistemic uncertainty of a predictor at $o$.

To overcome this issue with aleatoric uncertainty in RC-GVF, we propose to train an *ensemble* of predictors and utilise the variance across their predictions as a multiplicative factor on the prediction error. Concretely, we train $K$ predictors $\hat{v}_{\pi,z}^k, k \in \{1, 2 \ldots K\}$. The prediction target for each member of the ensemble is the same $\lambda$-pseudo-return (Equation 4) by using bootstrapped values from that member's predictions. Using the ensemble of predictors, the intrinsic reward is given by

$$R_i(o_t) = \sum_{j=1}^{d} \left( \mathbb{E}[L_{\text{TD}}^k(h_t)] \odot \mathbb{V}[\hat{v}_{\pi,z_j}^k(h_t)] \right)_j \tag{6}$$

$$= \sum_{j=1}^{d} \left[ \frac{1}{K} \sum_{k=1}^{K} \left( G_t^{z_j} - \hat{v}_{\pi,z_j}^k(h_t) \right)^2 \right] \cdot \left[ \frac{1}{K-1} \sum_{k=1}^{K} \left( \bar{v}_{\pi,z_j}(h_t) - \hat{v}_{\pi,z_j}^k(h_t) \right)^2 \right], \tag{7}$$

where $\odot$ corresponds to element-wise multiplication. In this formulation, even when prediction error remains, the exploration bonus will vanish as the predictors converge to the same expectation.

## 3.3 Effective horizon and relation to random network distillation

The effective horizon over which predictions are considered depends on the choice of the discount factor $\gamma_z$. It can be shown that for a horizon $H_{\gamma_z,\varepsilon} \in \mathbb{R}$ satisfying

$$H_{\gamma_z,\varepsilon} \geq \frac{\log\left(\frac{1}{\varepsilon(1-\gamma_z)}\right)}{\log\left(\frac{1}{\gamma_z}\right)}, \tag{8}$$

the discounted sum of (pseudo) rewards beyond this horizon is bounded by $\varepsilon$ [28, 30].

From Equation 4 we can see that for the special case of $\gamma_z = 0$ and any choice of $\lambda_z$, we have $G_t^z = Z_{t+1} = Z_\phi(o_t)$. Note that in this case the TD-error between prediction and target is equivalent to the intrinsic reward provided by random network distillation (RND; Equation 2). Since the prediction targets are now deterministic, there is no problem with aleatoric uncertainty, and a single predictor suffices (as opposed to an ensemble).

# 4 Related work

**Exploration** In tabular RL, explicit *counts* for states or state-action pairs can be maintained to enable exploration and achieve efficient learning [61, 4]. Count-based bonuses have been scaled to larger state spaces through the use of density models, which provide pseudo-counts [7, 41]. State novelty bonuses are similarly inspired heuristics that provide a bonus based on estimated novelty of the visited state [10].

Perhaps the simplest way of implementing exploration through *'curiosity'* is to reward the policy for encountering transitions to states that surprise a learning predictor [52]. This is suitable for deterministic environments but it suffers from the 'noisy TV problem' in stochastic environments [56, 9]. To overcome this limitation, intrinsic rewards have been defined by *learning progress* (the first derivative of the prediction error) [51], *information gain* [60, 22], or *compression progress* [55]. Also compare such bonuses to pseudo-counts [7]. Alternatively, it is possible to mitigate sensitivity to noise by measuring prediction errors in a latent space [53, 43].

Our method is different from state count and novelty approaches in that it models temporally extended values beyond the current observation. It includes a state-novelty measure, RND [10], as a special case when the GVF discount factors are zero. Unlike most artificial curiosity approaches, RC-GVF does not model the entire environment dynamics, in line with the idea of conducting temporally extended experiments whose outcomes are abstract summaries (represented in latent variables) of long observation sequences [54]. We mitigate sensitivity to aleatoric uncertainty by measuring the disagreement of an *ensemble* of predictors, inspired by prior work (for single-step prediction models) [59, 44]. Ensembles for exploration have also been explored in the context of posterior sampling [62]. Bootstrapped-DQN [38] uses an ensemble of Q-value functions for a model-free interpretation of posterior sampling [39]. Unlike in our approach, these value functions are restricted to the original task rewards, and can not capture arbitrary pseudo-rewards. Recent works suggest incorporating model-based planning to directly optimise for long-term novelty [63, 58, 31]. Our intrinsic reward–which is generated by considering multiple future steps–could motivate approaches for optimising long-term novelty without explicit rollout-based planning.

**General Value Functions** Sutton et al. [67] proposed general value functions (GVFs) as an approach to represent predictive knowledge about the world. Closely related is the vector-valued adaptive critic [50] which predicts and controls cumulative values of special input vectors. Several works have studied the impact of using auxiliary value functions to improve representation learning in RL [24, 8, 35]. Further, it was previously found that GVFs of randomly generated pseudo-rewards can also be useful for shaping and learning representations [34, 78] (see also related work on forecasts [47], TD-networks [66], and predictive state representations [32]). This can be viewed as 'answering' questions about cumulative quantities of interest under a particular policy $\pi$ and discount factor $\gamma_z$. In our implementation, the predictor is an entirely separate network and thus GVF prediction can not be used to improve the agent's internal representation. Instead, we use the GVF prediction to derive a useful signal to guide exploration.

The discovery/selection of useful GVF 'questions' is an open problem [48]. Existing approaches learn questions related to optimizing performance on the main task [73, 27]. However, such an approach might not be suitable for sparse-reward environments. In our approach, we side-step the GVF discovery problem through the use of randomly generated pseudo-rewards and on-policy predictions.

# 5 Experiments

In this section we present an empirical evaluation of RC-GVF.[2] First, we consider a hard exploration diabolical lock problem from the literature [36] and demonstrate the benefits of predicting beyond the immediate observation. We then proceed to experiments with the standard Minigrid suite of partially observable and procedurally generated environments, demonstrating how RC-GVF is able to outperform several strong baselines [18, 77].

---

[2]Code is available at `https://github.com/Aditya-Ramesh-10/exploring-through-rcgvf`.

## 5.1 Diabolical locks

The diabolical lock problem [36] is considered a difficult environment for exploration due to noisy high-dimensional observations, stochastic dynamics and misleading rewards that deter the agent from finding the sparse optimal reward at the end of the lock. In the literature, RND is often shown to fail on these types of environments due to only considering the immediate future [36]. Due to RC-GVF's policy-conditioned predictions of future quantities, this task is instructive to demonstrate the qualitative advantages of our method and how this results in markedly different performance.

**Environment details** The environment consists of states organised in 3 rows $\{a, b, c\}$ and $H$ columns from $\{1, 2, \ldots H\}$ (see Figure 6 in Appendix A.1). In each episode, the agent is initialised in one of two possible starting states, either at $a_1$ or $b_1$. At each state the agent has $L$ available actions, only one of which is 'good' and transitions to a 'good' state in the next column with equal probability. Taking the good action gives a negative (anti-shaped) reward of

Table 1: Mean over 10 seeds and (min, max) of the farthest column reached (of $H = 100$) at different points on the diabolical lock problem.

| Frames | 5M | 10M | 20M |
|---|---|---|---|
| RC-GVF | 61 (47, 81) | 94 (83, 100) | 100 (100, 100) |
| RND | 28 (21, 41) | 37 (27, 47) | 46 (33, 58) |

$-1/H$, except at the end of the lock (states $a_H$ or $b_H$), where the sparse optimal reward of 10 is received. The remaining $L - 1$ actions at any state are 'bad', causing a deterministic transition to the 'dead' row $c$ at the next column (zero reward). All actions from a dead state lead to the dead state in the next row, and hence the optimal policy is to take the respective good actions in each state. The good action at each state is assigned randomly as part of the MDP specification. The agent never observes the state directly and only has access to a high-dimensional noisy observation (details in Appendix A.1).

In our experiment, we consider a problem with horizon $H = 100$, $L = 10$, and observation noise $\sigma_o = 0.1$. As discussed by Misra et al. [36], this exploration problem is hard because the probability of reaching the optimal reward through a uniformly random policy is $L^{-H}$ ($10^{-100}$ in our instance). Further, the stochastic transitions from good states prevent a solution that relies on memorising a state-independent successful sequence of $H$ actions independent of the state.

**Implementation** We use Proximal Policy Optimization (PPO) [57] in an actor-critic framework as the base agent and use 128 pseudo-rewards for RND and RC-GVF. For RC-GVF we set $\gamma_z = 0.6$ and use two prediction heads in the ensemble. The agent is trained to maximize the expected sum of a weighted combination of intrinsic and extrinsic rewards. Further details are in Appendix A.1.

**Results** Table 1 compares RND to RC-GVF in this environment. It can be seen how RC-GVF succeeds in completing the lock on all runs, while the best run of RND only reaches $60\%$ of the lock (and does not complete it). Indeed, once the agent takes an incorrect action and 'falls' into the bottom row, the future consequences (and thus, the GVFs) become highly predictable. By staying alive, the agent will encounter new observations and unpredictable transitions which appear interesting. Further experiments with additional baselines are presented in Appendix D. This result is illustrative of RC-GVFs exploration capabilities as we will see next.

## 5.2 MiniGrid

We evaluate RC-GVF on procedurally generated environments from MiniGrid [13], which is a standard benchmark in the deep reinforcement learning literature for exploration [46, 11, 18, 77, 76, 42]. Exploration in these environments is challenging due to partial observability, extremely sparse rewards, and the procedural generation of mazes and objects.

**Environment Details** Broadly speaking, we will consider three classes of environments organised according to several difficulty levels. First, we study *MultiRoom-N7-S8* and *MultiRoom-N12-S10*, which are navigation tasks in a maze with seven and twelve rooms respectively. Next, we consider two levels of difficulty in *KeyCorridor-S4R3* and *KeyCorridor-S5R3*. Here, the agent needs to pick up a ball behind a locked door. Finally, we examine the *ObstructedMaze* set of environments where the agent has a similar task of picking up a ball behind a locked door, but keys are hidden in boxes (*ObstructedMaze-2Dlh*), and doors can be obstructed (*ObstructedMaze-2Dlhb*). The observations

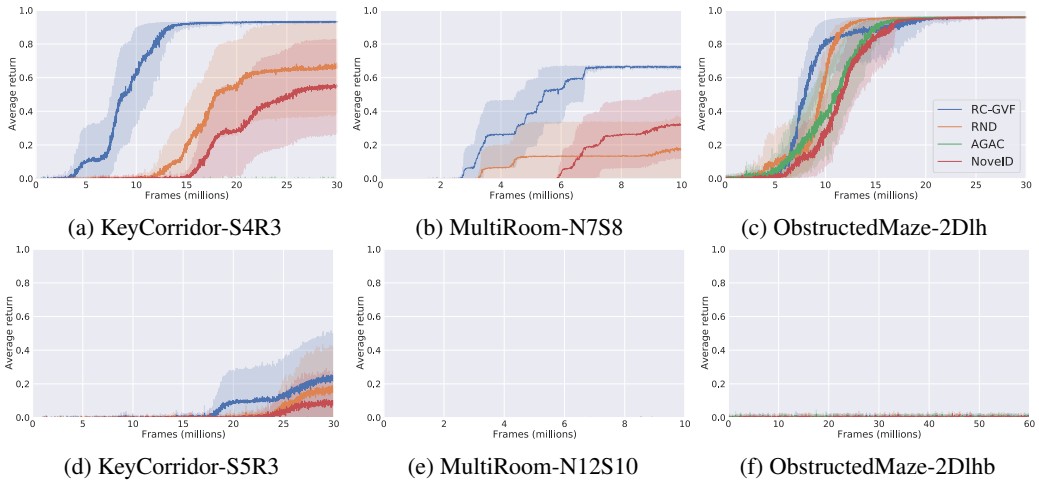

Figure 2: Average return of RC-GVF , RND and other baselines on the selected Minigrid environments (without ground-truth episodic counts for baselines). RC-GVF outperforms baselines in the absence of episodic counts. Some environments prove challenging for all approaches in this setting.

corresponds to an egocentric view of the cells in the front of the agent. Further details of the MiniGrid environments are available in Appendix A.2.

**Implementation**  We compare RC-GVF to RND [10], AGAC [18], and NovelD [77]. We use Proximal Policy Optimization (PPO) [57] as our base agent for all approaches. The agent is trained to maximize the expected sum of a weighted combination of intrinsic and extrinsic rewards. At each time step $t$, the agent receives a reward $R_t = R_e(s_t) + \beta R_i(o_t)$, where $\beta \in \mathbb{R}_+$ balances the contribution of the extrinsic $R_e(s_t)$ and intrinsic $R_i(o_t)$ rewards. For RC-GVF we set $\gamma_z = 0.6$, use two prediction heads in the ensemble of predictors and use 128 pseudo-rewards (same as for RND). Other important hyper-parameters, such as the intrinsic reward coefficient ($\beta$), entropy coefficient, and learning rate of the predictor are obtained via an extensive hyper-parameter search for all baselines (see Appendices C.2 and C.3 for details, including on our implementation of baselines.).

We present results averaged over 10 independent runs for each approach in every figure (solid line). Unless mentioned otherwise, the shading indicates 95% bootstrapped confidence intervals.

### 5.2.1 Egocentric observations

We will first consider the usual setting where the agent receives access to the egocentric observations from the environment. Importantly, we do not provide agents with access to privileged information about the environment in the form of so-called 'episodic counts'. This is different from several recent approaches that have been applied to MiniGrid, which incorporate these values as part of their intrinsic reward [46, 18, 77].

The use of episodic counts obtained through the simulator is problematic for solving exploration problems in a partially observable setting, since it tells the agent precisely how frequently it has been in each state. Indeed, the use of episodic counts *alone* can be sufficient for exploration in MiniGrid (Figure 7 in Appendix E.1). Meanwhile, there exists no good method for obtaining episodic counts in the absence of a simulator, as it is difficult to accurately estimate pseudo-counts from partial observations (eg. as in Figure 1). Details about the exact usage of episodic counts by baselines in their intrinsic reward formulation are provided in Appendix C.2.

In Figure 2 we compare RC-GVF, RND, AGAC, and NovelD in the absence of episodic counts, while using egocentric observations. It can be seen that RC-GVF explores faster and succeeds more often than RND and the other baselines in the KeyCorridor-S4R3 environment and MultiRoom-N7-S8 environment. In comparison to RND and NovelD that utilise prediction errors in immediate random embeddings, it appears that RC-GVF benefits from extended horizon predictions of random pseudo-rewards. Meanwhile, AGAC only succeeds in ObstructedMaze-2Dlh, where all baselines perform similarly well. Finally, we observe that all approaches struggle with the most difficult level of considered environments (KeyCorridor-S5R3, MultiRoom-N12-S10 and ObstructedMaze-2Dlhb).

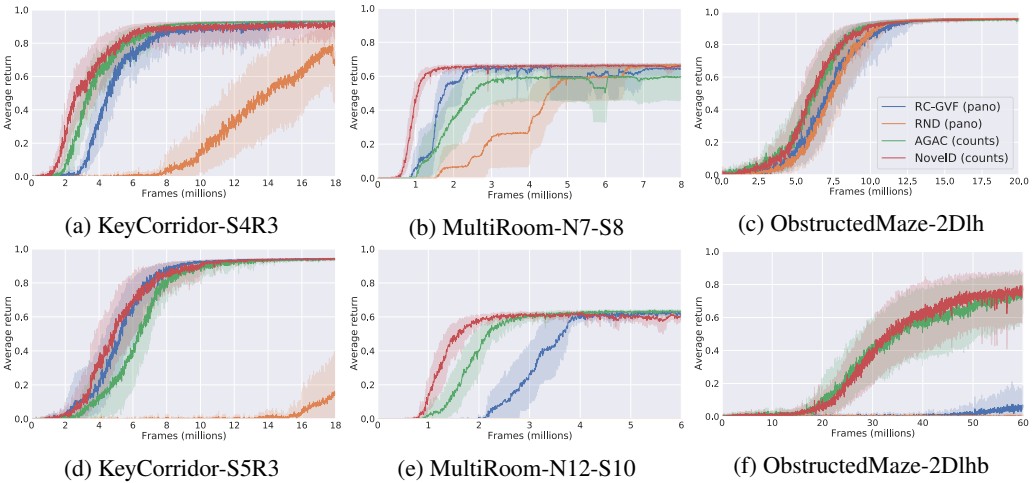

Figure 3: Average return of RC-GVF and RND with panoramic observations. The performance of baselines (AGAC and NovelD) improves significantly with the inclusion of the episodic count component while RC-GVF (using panoramic observations) achieves similar performance without.

### 5.2.2 Panoramic observations

We now demonstrate how using episodic counts from the simulator is crucial to the success of the baselines. In Figure 3 it can be seen how AGAC and NovelD improve with episodic counts, and also solve the more difficult problems (KeyCorridor-S5R3, MultiRoom-N12-S10 and ObstructedMaze-2Dlhb).

**Panoramic observations**   In order to increase performance of RC-GVF further, without introducing episodic counts, we investigate the use of augmented panoramic views as previously explored by Parisi et al. [42][3]. In the MiniGrid environments, the observation changes almost in its entirety when the agent changes the direction it faces. In contrast, moving to the next cell leads to fewer sudden changes in the observation.

The lack of gradual changes of egocentric observations through rotations is a consequence of having four discrete angles for orientation $(0°, 90°, 180°, 270°)$, which may be an unrealistic depiction of how rotations affect observations for agents situated in the real world. As a consequence, prediction errors are dominated by predictions of the outcomes of turning (rather than other actions). To address this, prior work proposed to make the agent's observations invariant to rotation by augmenting the observation with all directional observations [42]. In effect, this assumes that the agent rotates $360°$ after moving to a cell, or alternatively that it is equipped with additional sensors on its sides and back. Similar to Parisi et al. [42], we will consider panoramic views only for generating intrinsic rewards, and egocentric observations for the base PPO agent.

In Figure 3, we demonstrate how RC-GVF with panoramic views and without privileged information about the underlying state of the MDP can solve harder problems and is competitive with the baselines that use episodic counts in five out of the six settings. We see that RC-GVF with panoramic observations comfortably succeeds in the KeyCorridor-S5R3 and MultiRoom-N12-S10 environments where it was previously unsuccessful. RC-GVF did not succeed in solving the harder ObstructedMaze-2Dlhb within the given frames despite the inclusion of panoramic views. Figure 3 shows that the use of panoramic views does not make the exploration problem trivial; RND also uses panoramic views (but no episodic counts). It improves with panoramic views on KeyCorridor-S4R3 and MultiRoom-N7S8 but struggles with the harder instances.

### 5.3 Analysis

**Changing the temporal prediction horizon**   We study the effect of changing the temporal prediction horizon by evaluating RC-GVF with different discount factors. Recall from Equation 8

---

[3]We verified in a preliminary experiment that incorporating episodic counts for RC-GVF similarly improves performance (see Figure 8 in Appendix E.2). However, since we believe that this constitutes privileged information that is not readily accessible outside of a simulator, we will not explore this direction further.

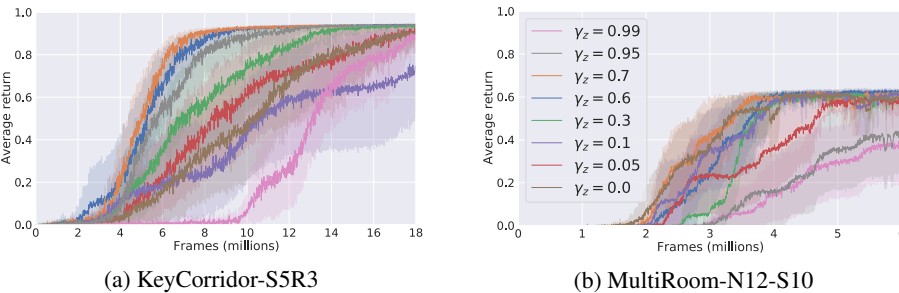

(a) KeyCorridor-S5R3                    (b) MultiRoom-N12-S10

Figure 4: Analysis reveals that: (a) larger discount factors (except $0.99$) are useful in KeyCorridor-S5R3 (b) low and intermediate discount factors perform well in MultiRoom-N12-S10.

how the discount factor can me mapped to the prediction horizon, which lets us estimate a lower bound on the horizon of about $11$ time steps for $\gamma_z = 0.6$ with $\epsilon = 0.01$. In Figure 4 it can be seen how a larger prediction horizon is desirable for KeyCorridorS5R3. Among the higher discount factors, $0.95$ is a good choice for this environment, but not $0.99$. Compared to our default value of $\gamma_z = 0.6$, a short horizon using $0.3$ reduces performance further, while discount factors closest to $0$ yield worse performance. In contrast, for MultiRoom-N12-S10, we find that low and intermediate values of $\gamma_z$ perform well. The high GVF discount factors of $0.99$ and $0.95$ perform worse. Comparing $\gamma_z = 0$ to RND shows how the use of recurrent networks and the variance term derived from the ensemble of predictors contributes most to the performance of RC-GVF in this domain.

**Ablation**   To better understand which of these components are contributing most to the success of RC-GVF, we consider the following additional ablations: (1) RC-GVF ($\gamma_z = 0$) without the variance term in Equation 6, and (2) RC-GVF ($\gamma_z = 0$) without the recurrent predictor. Figure 5a compares these variations to RC-GVF ($\gamma_z = 0$), RC-GVF ($\gamma_z = 0.6$), and RND. It can be seen that the inclusion of the history-conditioned recurrent predictor and the variance term individually contribute to the improved performance of RC-GVF.

Using the history conditioned recurrent predictor sharply improves the performance of RC-GVF with $\gamma_z = 0$. We hypothesise that this effect arises from the previous pseudo-rewards available to the predictor, which might enable better predictions on input observations it was not explicitly trained on (which is often the case in procedurally generated environments). In support of this hypothesis, we observed that using a recurrent predictor with solely observations as inputs does not produce such an improvement (Figure 9 in Appendix E.3).

In Figure 5b, we present ablations for $\gamma_z = 0.6$. We consider RC-GVF without the variance term in Equation 6, and without the recurrent predictor. Removing either the disagreement term or the recurrent predictor worsens the performance of RC-GVF with $\gamma_z = 0.6$. Interestingly, unlike as was observed for $\gamma_z = 0$, here we find that RC-GVF with only disagreement (no recurrent predictor) performs better than RC-GVF with only recurrent predictor (no disagreement). This further emphasises the importance of handling the aleatoric uncertainty for non-zero GVF discounts.

**Increasing the ensemble size**   We conduct experiments with larger ensemble sizes in two MiniGrid environments. We obtain comparable results for ensemble sizes of 2, 4, 6 and 8 predictors in Figure 10 in Appendix E.4, indicating that two member ensembles usually suffice for these problems.

## 6   Limitations and Societal Impact

**Limitations**   The incorporation of the current policy into the prediction task of GVF prediction can have benefits with regards to behavioural exploration. However, it may also encourage over-exploration due to the same state appearing 'interesting' under different policies. Possible mitigation strategies include off-policy variants or explicitly policy-conditioned (general) value functions [17, 20].

In some situations it may be difficult to ascertain what is a good choice of the GVF discount factor $\gamma_z$, which effectively determines the prediction horizon length, and can be influenced by optimisation effects under function approximation [72]. The best choice may be problem-dependent and could

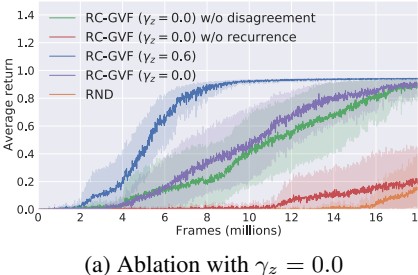
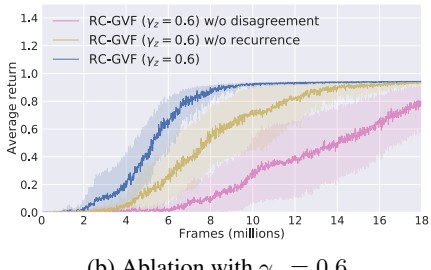

(a) Ablation with $\gamma_z = 0.0$                 (b) Ablation with $\gamma_z = 0.6$

Figure 5: Ablation of RC-GVF components in the KeyCorridor-S5R3 environment (a) the variance term, recurrent predictor and discount factor contribute most to the performance of RC-GVF (b) the variance term appears to be more important in the case of $\gamma_z = 0.6$.

further benefit from dynamic adjustment during learning [5, 37]. Nonetheless, our results in Figures 2 and 3 indicate that a default of $\gamma_z = 0.6$ works well in many of the considered settings.

Several other intrinsic reward formulations using RC-GVF are possible (eg. based on information/prediction gain, learning progress, or only the disagreement term), which we did not investigate. Our specific formulation in Equation 6 was motivated by the prediction error in RND, which does not separate the aleatoric uncertainty when $\gamma_z > 0$. Starting with the prediction error term also allowed us to closely study the connection between RND and RC-GVF.

A general limitation of curiosity-based exploration for task-specific RL is that more information is gathered than is necessary to solve the given task [40, 33]. Indeed, the space of potential GVFs is enormous. In the case of RC-GVF one could aim at focusing it by integrating task-specific information [68] or by considering predictions using a smaller set of policies that adequately cover the state space [1].

**Societal impact**    This paper does not focus on real-world applications of exploration in RL and thus does not have a direct societal impact. In applications, advanced exploration methods may lead to unexpected policy behaviour, which could be mitigated by incorporating safety constraints [19].

## 7   Conclusion

We introduced *random curiosity with general value functions (RC-GVF)*, an exploration method that intrinsically rewards a reinforcement learning agent based on errors and uncertainty in predicting random transformations of observation sequences generated through the agent's actions. Unlike state-novelty approaches, ours takes into account multi-step future behaviours of policy and environment. Unlike commonly used artificial curiosity approaches, RC-GVF does not rely on model rollouts and does not need to predict all details of future observations.

On the diabolical lock problem [36] and the MiniGrid environments [13] we demonstrate that predicting abstract quantities of extended time intervals can improve exploration in POMDPs. Compared to recent methods such as AGAC [18] and NovelD [77] that rely on privileged episodic state-visitation counts in MiniGrid, our RC-GVF achieves competitive results even when using only panoramic observations. Importantly, in the natural setting where episodic counts are not available, RC-GVF significantly outperforms all baseline methods.

Our approach can be generalized by moving beyond random pseudo-rewards, considering general value functions under a set of different policies [1], and introducing time or state-dependent discounting. This offers exciting avenues for future research.

## Acknowledgments and Disclosure of Funding

We would like to thank Kenny Young, Francesco Faccio, Anand Gopalakrishnan, and Dylan Ashley for valuable comments. This research was supported by the ERC Advanced Grant (742870), the Swiss National Science Foundation grant (200021_192356), and by the Swiss National Supercomputing Centre (CSCS projects s1090 and s1127).

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
