# A Environments

In the following sections we describe the environments used in our experiments.

## A.1 Diabolical lock

The diabolical lock problem [36] consists of states organised in 3 rows $\{a, b, c\}$ and $H$ columns from $\{1, 2, \ldots H\}$ (see Figure 6). In each episode, the agent is initialised in one of two possible starting states, either at $a_1$ or $b_1$ with equal probability. Each episode lasts for $H$ time steps.

**Transition dynamics**  In each state the agent has $L$ available actions, only one of which is 'good' and transitions to a 'good' state (green) in the next column with equal probability. For instance, the good action in state $a_t$ leads to $a_{t+1}$ or $b_{t+1}$ with equal probability. Taking the good action gives a negative (anti-shaped) reward of $-1/H$, except at the end of the lock (states $a_H$ or $b_H$), where the sparse optimal reward of 10 is received. The remaining $L - 1$ actions in any state are 'bad,' causing a deterministic transition to the 'dead' row $c$ (red) at the next column (with zero reward). The agent remains in the 'dead' row $c$ till the end of the episode, i.e., all actions from $c_t$ transition to $c_{t+1}$ with zero reward. The good action in each state is assigned uniformly at random from the $L$ available actions upon initialising the environment. The good action in every state remains fixed across episodes.

**Noisy observations**  The diabolical lock environment provides a high-dimensional noisy observation based on the environmental state. The observation is obtained by concatenating one-hot vectors encoding the row and column, adding independent Gaussian noise with variance $\sigma_o^2$ to each component of the vector, padding this vector with zeroes to the nearest higher power of two and then applying a Hadamard rotation matrix.

In our experiments, we select $H = 100$, $L = 10$, and $\sigma_o = 0.1$, which is similar to the configuration used by Misra et al. [36]. We were unable to find a publicly available implementation of the environment introduced by Misra et al. [36] and therefore used our own (no license).

## A.2 MiniGrid

The MiniGrid [13] (Apache License 2.0) set of environments are partially observable, procedurally generated, and provide a sparse reward upon successful completion of the task.

The agent receives a $7 \times 7$ egocentric view of its surroundings, which is a $7 \times 7 \times 3$ integer encoding that describes the contents of the visible grid cells. The agent cannot see behind walls or closed doors.

For the study with *panoramic observations*, we follow the implementation of Parisi et al. [42] where a rotation invariant view is created by concatenating the four observations from when the agent faces north, east, south and west to create a $7 \times 7 \times 12$ panoramic observation.

At every step, the agent has seven available actions: turn left, turn right, move forward, pick up, drop, toggle (used to open doors), and done.

A new configuration of the environment is generated for each episode. Episodes finish when the agent has interacted with the environment for the maximum number of steps permitted, or if the agent successfully completes the task, in which case it also receives a non-zero reward depending on the number of steps taken to complete the task.

**KeyCorridor**  The agent has to pick up a ball behind a locked door. To unlock the door, the agent must first find the key, which is also hidden behind closed (but unlocked) doors. We consider two levels of difficulty in this class of environments. The KeyCorridor-S5R3 environment poses a harder exploratory challenge due to larger rooms and hallways in comparison to KeyCorridor-S4R3.

**ObstructedMaze**  Like with the KeyCorridor environments, the goal of the agent is to pick up a ball behind one of the locked doors. We again consider two levels of difficulty in this class of environments. In the ObstructedMaze-2Dlh environment the keys are hidden inside chests. In ObstructedMaze-2Dlhb the keys are hidden inside chests and the doors are blocked by obstacles that need to be displaced.

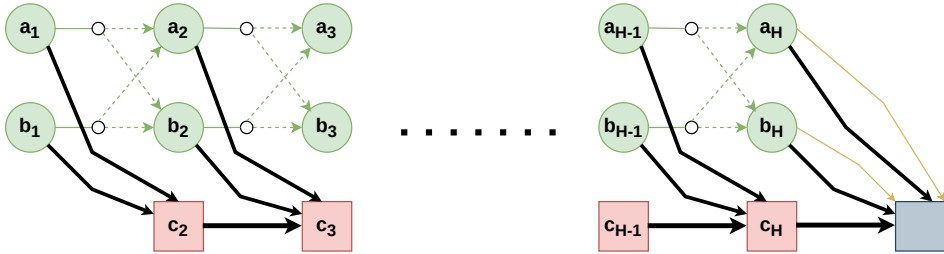

Figure 6: Based on Figure 5 from Misra et al. [36]. The diabolical lock environment is organised in 3 rows $\{a, b, c\}$ and $H$ columns. The green states of rows $a$ and $b$ correspond to the 'good' states. The red states of row $c$ correspond to the 'dead' states. The grey square denotes the terminal state. The black arrows represent deterministic transitions to the 'dead' row of the next column. The green dotted arrows denote the equally likely transitions to one of two good states by taking the 'good' action. The yellow arrows mark the final transitions which provide the optimal sparse reward upon termination.

**MultiRoom**    The goal of the agent is to reach the goal state in the farthest room. The notation N12-S10 signifies that the environment consists of 12 rooms with a maximum size of 10. These environments (MultiRoom-N7-S8 and MultiRoom-N12-S10) are not available in the default set of MiniGrid environments. We borrow the implementation of these environments from previous work [46](Creative Commons Attribution-NonCommercial 4.0 International Public License).

## B    Implementation details for diabolical lock experiments

This section presents the implementation details for our experiments with the diabolical lock environment. In the following, we describe the architectures and hyperparameters of our actor, critic, pseudo-reward generator, and ensemble of predictors used in these experiments.

### B.1    Neural network architectures

**Base agent**    We use the PPO [57] implementation from the `rl-starter-files` repository [74]. The actor and critic share a multi-layer perceptron (MLP) of three layers with 256 units and ReLU non-linearities. Separate linear heads for the actor and critic operate on the representation provided by the shared architecture.

**Pseudo-reward generator**    The randomly initialised fixed network for generating pseudo-rewards is a two layer MLP with ReLU non-linearities in the hidden layer of 128 units. The linear output layer provides a vector of 128 pseudo-rewards.

**Predictor networks**    The predictor network for RND has the same architecture as the pseudo-reward generator.

The predictor network for RC-GVF is an MLP (no recurrence here) with wider layers and one extra layer compared to the pseudo-reward generating network. Specifically, a predictor consists of a network of three layers with 256 units each with ReLU non-linearities in the two hidden layers. We use an ensemble of $K = 2$ predictors in the case of RC-GVF. In these experiments with the diabolical lock, the individual predictors in the ensemble are completely separate neural networks which do not share parameters.

### B.2    Grid search and hyperparameters

Hyperparameters of the base PPO agent are provided in Table 2. These were selected based on previous work [36] and apply to our method and RND.

Utilising an intrinsic reward introduces additional hyperparameters. We conducted a grid-search to select the values for hyperparameters introduced by RND and RC-GVF. The best hyperparameter configuration was selected by evaluating average lock completion for each candidate in the grid

Table 2: PPO hyperparameters for diabolical lock experiments.

| Hyperparameter | Value |
|---|---|
| Discount factor | 0.99 |
| GAE lambda | 0.95 |
| Optimization epochs | 5 |
| Learning rate ($\eta$) | 0.0005 |
| Clipping epsilon | 0.2 |
| Rollout length (frames per process) | $H = 100$ |
| Batch size | 256 |
| Parallel processes | 16 |
| Entropy coefficient | 0.01 |
| Value loss coefficient | 0.5 |

across 5 seeds. In Table 1, we report the final results with 10 independent runs of the selected hyperparameters.

For RND, the hyperparameters include the intrinsic coefficient $\beta$ and the learning rate of the predictor $\eta_p$. The search candidates and the final selected values are described in Table 3.

Table 3: RND specific hyperparameters for diabolical lock experiments.

| Hyperparameter | Candidates | Selected |
|---|---|---|
| Predictor learning rate $\eta_p$ | $\{\eta, 0.75\eta, 0.5\eta, 0.25\eta, 0.1\eta\}$ | $0.25\eta\ (\eta = 0.0005)$ |
| Intrinsic reward coefficient $\beta$ | $\{4.0, 2.0, 1.0, 0.5, 0.25\}$ | 0.5 |

In comparison to RND, RC-GVF introduces additional hyperparameters such as the GVF discount factor $\gamma_z$, the associated bias-variance tradeoff parameter for the TD-targets $\lambda_z$, and the number of predictors in the ensemble. We observed good results with our default values of $\gamma_z = 0.6$ and an ensemble of 2 predictors. The search candidates and the final selected values for RC-GVF are described in Table 4.

Table 4: RC-GVF specific hyperparameters for diabolical lock experiments.

| Hyperparameter | Candidates | Selected |
|---|---|---|
| Number of predictors | $\{2\}$ | 2 |
| Predictor learning rate $\eta_p$ | $\{\eta, 0.5\eta, 0.25\eta\}$ | $0.5\eta\ (\eta = 0.0005)$ |
| Intrinsic reward coefficient $\beta$ | $\{4.0, 2.0, 1.0, 0.5\}$ | 2.0 |
| GVF discount factor $\gamma_z$ | $\{0.6\}$ | 0.6 |

The learning rates of the base agent and predictor are linearly annealed to zero over 100 million frames. We used the Adam optimiser [29] for the PPO agent and the predictor in all experiments.

## C   Implementation details for Minigrid experiments

This section presents the implementation details for our experiments with the MiniGrid environments.

### C.1   Neural network architectures

**Base agent**   We use the PPO implementation from the `rl-starter-files` [74] repository. We retain the same architecture for the actor and critic as provided in their implementation.

The actor and the critic share three convolution layers with 16, 32 and 64 output channels, with $2 \times 2$ kernels and a stride of 1. Each convolution layer is followed by a ReLU non-linearity. A $2 \times 2$ max-pooling is applied after the first convolution layer. The output from the convolution layers is flattened and provided to an LSTM [21] with a hidden dimension of 64. Separate multi-layer perceptron (MLP) heads for the actor and critic operate on the output of the LSTM to output the

action logits and the value. Each MLP has 64 hidden units with a hyperbolic tangent (tanh) activation function.

**Pseudo-reward generator** The pseudo-reward generator is implemented as a convolutional neural network whose output is flattened to provide a vector of pseudo-rewards $z_{t+1} \in \mathbb{R}^d$. It has a similar architecture to the convolutional layers used in the base agent.

Concretely, this network consists of three convolution layers, with 32, 64, and $d$ output channels respectively, with $2 \times 2$ kernels and a stride of 1. Here, $d$ is the number of pseudo-rewards, which we set as $d = 128$ for our experiments. ReLU non-linearity is applied after the first two convolution layers. Additionally, a $2 \times 2$ max-pooling is applied after the first convolution layer.

**Predictor networks** In the case of RND, the predictor network has the same architecture as the pseudo-reward generator.

For RC-GVF, we have an ensemble of predictors with $K$ members. An ensemble of $K$ two layer MLP heads (with 256 units and ReLU non-linearity) operate on a common representation of 128 dimensions generated by a shared recurrent (LSTM) core. The LSTM operates on histories of observations, actions, and pseudo-rewards.

The observation at time step $o_t$ is fed through three convolution layers with 32, 64 and 128 output channels, with $2 \times 2$ kernels and a stride of 1. Each convolution layer is followed by a ReLU non-linearity. A $2 \times 2$ max-pooling is applied after the first convolution layer. The output from the convolution layers is flattened and provided to a fully connected layer which provides a 64 dimensional representation of the input observation.

The action at the previous time step $a_{t-1}$ (as a one-hot input) is transformed into a 32 dimensional vector by a fully connected layer. A fully connected layer is also applied to the previous pseudo-reward vector $z_t$ to obtain a 32 dimensional vector. The observation, action and pseudo-reward representations are concatenated into a 128 dimensional vector which is fed as input to the LSTM (with hidden dimension of 128).

### C.2 Details about baseline implementations

We employed the same base PPO agent (see Appendix C.1) for all approaches to maintain consistency. This means that our implementation of baselines deviate slightly from the original implementations. Nevertheless, we obtain comparable or better results than the reported ones on the considered environments. We describe the key details of our AGAC [18] and NovelD [77] implementations below.

**AGAC** The PPO implementation of AGAC utilised independent convolutional neural networks for the actor and critic with frame stacking rather than a shared recurrent neural network (for actor and critic) as used by us.

In our implementation, the policy predictor (or adversary) has the same architecture as the recurrent actor-critic (see Appendix C.1), but without the MLP head for the critic.

**NovelD** NovelD derives intrinsic rewards using RND as a novelty measure. The neural network architecture details for NovelD therefore follow those described for RND (see Appendix C.1).

A major difference is that the original implementation of NovelD used IMPALA [16] as the base agent instead of PPO.

Another deviation in our version of NovelD is that we do not use the *Episodic Restriction on Intrinsic Reward (ERIR)* multiplier for using episodic counts. The ERIR multiplier uses episodic counts from the simulator to ensure that intrinsic rewards are provided only when the agent visits a state for the first time in an episode. For consistency, we use episodic counts in a similar manner to RIDE [46] and AGAC [18], where the square root of episodic count divides the main intrinsic reward term. Our

NovelD intrinsic reward is given by[4]

$$R_i(o_t; s_t, o_{t+1}) = \frac{\max \left[ novelty(o_{t+1}) - \alpha \cdot novelty(o_t), 0 \right]}{\sqrt{N_e(s_t)}},$$ (9)

where $novelty(.)$ is measured as the RND prediction error (see Equation 2), $\alpha$ is a hyperparameter called the scaling factor, and $N_e(s_t)$ is the number of times the agent has been in that state in that episode.

**On the use of episodic counts**   Note that in our experiments the episodic counts refer to agent positions alone, following AGAC [18]. Other approaches have used different strategies to obtain episodic counts from the simulator. For example, RIDE [46] and NovelD [77] derive counts from fully observable views which provide count information based on agent position, direction or whether an object has been picked up etc.

### C.3   Grid search and selected hyperparameters

In this section we describe our hyperparameter selection strategy for the MiniGrid experiments.

Our implementation utilises the PPO implementation from the `rl-starter-files` [74] repository. The hyperparameters for the common PPO base agent were selected via preliminary experiments and are similar to the ones provided in the `rl-starter-files` repository. These are detailed in Table 5.

Table 5: PPO hyperparameters for MiniGrid experiments.

| Hyperparameter | Value |
|---|---|
| Discount factor | 0.99 |
| GAE lambda | 0.95 |
| Optimization epochs | 4 |
| Learning rate ($\eta$) | 0.0002 |
| Clipping epsilon | 0.2 |
| Rollout length (frames per process) | 128 |
| Batch size | 256 |
| Parallel processes | 16 |
| Value loss coefficient | 0.5 |
| Recurrence (for truncated BPTT [75]) | 4 |

We choose the hyperparameters introduced by each intrinsic reward approach via a grid search. We select separate hyperparameters for each environment type: MultiRoom, KeyCorridor and ObstructedMaze. To select hyperparameters, we evalaute each candidate configuration with 5 seeds on the simpler versions of the considered environments (MultiRoom-N7S8, KeyCorridor-S4R3 and ObstructedMaze-2Dlh) and select the configuration with the highest average return.

Final results are reported with separate independent runs on these environments (usually with 10 seeds) and also the harder versions of the considered environments (MultiRoom-N12S10, KeyCorridor-S5R3 and ObstructedMaze-2Dlhb).

All experiments used the Adam optimiser [29] for the PPO agent and the respective predictors.

Similar to previous approaches [46, 18], the learning rates are linearly annealed to zero while interacting with the environment during 10 million frames for the MultiRoom environments, 30 million frames for the KeyCorridor and ObstructedMaze-2Dlh environments, and 90 million frames for the ObstructedMaze-2Dlhb environment.

#### C.3.1   RC-GVF

We selected the GVF discount factor $\gamma_z = 0.6$ and the associated $\lambda$-return parameter $\lambda_z = 0.9$ as default values for all experiments. We found that $K = 2$ predictors in the ensemble worked

---

[4]We note that the authors of NovelD mention that replacing the ERIR term with scaling based on the square root of episodic counts still outperforms previous baselines (https://openreview.net/forum?id=CYUzpnOkFJp).

sufficiently well in preliminary experiments. We identified the entropy coefficient, intrinsic reward coefficient and learning rate of the predictor through a grid search. The candidate and selected values are described in Tables 6 and 7 for the setting with egocentric and panoramic observations respectively.

Table 6: RC-GVF specific hyperparameters for MiniGrid experiments with egocentric observations.

| Hyperparameter | Candidates | Egocentric observations | | |
| --- | --- | --- | --- | --- |
| | | MultiRoom | KeyCorridor | ObstructedMaze |
| Number of predictors | $\{2\}$ | 2 | 2 | 2 |
| GVF discount factor $\gamma_z$ | $\{0.6\}$ | 0.6 | 0.6 | 0.6 |
| GVF $\lambda_z$ (for TD-$\lambda$ targets) | $\{0.9\}$ | 0.9 | 0.9 | 0.9 |
| Predictor learning rate $\eta_p$ | $\{0.5\eta, 0.75\eta, \eta\}$ | $0.75\eta$ | $0.75\eta$ | $0.75\eta$ |
| Intrinsic reward coefficient $\beta$ | $\{0.1, 0.5, 1.0\}$ | 1.0 | 0.1 | 0.1 |
| Entropy coefficient | $\{0, 1e-5, 1e-4, 1e-3\}$ | 1e-5 | 1e-5 | 1e-4 |

Table 7: RC-GVF specific hyperparameters for MiniGrid experiments with panoramic observations.

| Hyperparameter | Candidates | Panoramic observations | | |
| --- | --- | --- | --- | --- |
| | | MultiRoom | KeyCorridor | ObstructedMaze |
| Number of predictors | $\{2\}$ | 2 | 2 | 2 |
| GVF discount factor $\gamma_z$ | $\{0.6\}$ | 0.6 | 0.6 | 0.6 |
| GVF $\lambda_z$ (for TD-$\lambda$ targets) | $\{0.9\}$ | 0.9 | 0.9 | 0.9 |
| Predictor learning rate $\eta_p$ | $\{0.5\eta, 0.75\eta, \eta\}$ | $0.5\eta$ | $0.75\eta$ | $0.5\eta$ |
| Intrinsic reward coefficient $\beta$ | $\{0.1, 0.5, 1.0\}$ | 0.1 | 1.0 | 0.1 |
| Entropy coefficient | $\{0, 1e-5, 1e-4, 1e-3\}$ | 1e-5 | 1e-5 | 1e-4 |

### C.3.2 RND

Tables 8 and 9 describe the search grid and selected hyperparameters for RND with egocentric and panoramic observations respectively.

Table 8: RND specific hyperparameters for MiniGrid experiments with egocentric observations.

| Hyperparameter | Candidates | Egocentric observations | | |
| --- | --- | --- | --- | --- |
| | | MultiRoom | KeyCorridor | ObstructedMaze |
| Predictor learning rate $\eta_p$ | $\{0.5\eta, 0.75\eta, \eta\}$ | $0.75\eta$ | $0.75\eta$ | $0.75\eta$ |
| Intrinsic reward coefficient $\beta$ | $\{0.0001, 0.0005, 0.001, 0.005\}$ | 0.0005 | 0.0005 | 0.0005 |
| Entropy coefficient | $\{0, 1e-5, 1e-4, 1e-3\}$ | 0 | 1e-5 | 1e-4 |

Table 9: RND specific hyperparameters for MiniGrid experiments with panoramic observations.

| Hyperparameter | Candidates | Panoramic observations | | |
| --- | --- | --- | --- | --- |
| | | MultiRoom | KeyCorridor | ObstructedMaze |
| Predictor learning rate $\eta_p$ | $\{0.5\eta, 0.75\eta, \eta\}$ | $0.75\eta$ | $0.75\eta$ | $0.75\eta$ |
| Intrinsic reward coefficient $\beta$ | $\{0.0001, 0.0005, 0.001, 0.005\}$ | 0.0005 | 0.001 | 0.0001 |
| Entropy coefficient | $\{0, 1e-5, 1e-4, 1e-3\}$ | 1e-5 | 1e-5 | 1e-4 |

### C.3.3 NovelD

NovelD relies on RND as a novelty measure and therefore has similar hyperparameters. We identified the entropy coefficient, intrinsic coefficient and learning rate of the RND predictor through a grid search.

A new hyperparameter introduced by NovelD is the scaling factor ($\alpha$ in Equation 9) which was selected as 0.5 in previous work [77]. We also found the value of 0.5 to work well in our preliminary experiments and did not search over other values for this hyperparameter.

Tables 10 and 11 describe the search grid and selected hyperparameters for NovelD.

Table 10: NovelD specific hyperparameters for MiniGrid experiments with egocentric observations (without counts).

| Hyperparameter | Candidates | Without episodic counts | | |
| --- | --- | --- | --- | --- |
| | | MultiRoom | KeyCorridor | ObstructedMaze |
| Scaling factor $\alpha$ | $\{0.5\}$ | 0.5 | 0.5 | 0.5 |
| Predictor learning rate $\eta_p$ | $\{0.5\eta, 0.75\eta, \eta\}$ | $0.5\eta$ | $0.75\eta$ | $0.75\eta$ |
| Intrinsic reward coefficient | $\{0.0005, 0.001, 0.005, 0.01\}$ | 0.01 | 0.005 | 0.001 |
| Entropy coefficient | $\{0, 1e-5, 1e-4, 1e-3\}$ | 0 | 1e-5 | 1e-4 |

Table 11: NovelD specific hyperparameters for MiniGrid experiments with episodic counts.

| Hyperparameter | Candidates | With episodic counts | | |
| --- | --- | --- | --- | --- |
| | | MultiRoom | KeyCorridor | ObstructedMaze |
| Scaling factor $\alpha$ | $\{0.5\}$ | 0.5 | 0.5 | 0.5 |
| Predictor learning rate $\eta_p$ | $\{0.5\eta, 0.75\eta, \eta\}$ | $0.75\eta$ | $0.75\eta$ | $0.75\eta$ |
| Intrinsic reward coefficient $\beta$ | $\{0.005, 0.01, 0.05, 0.1\}$ | 0.05 | 0.01 | 0.01 |
| Entropy coefficient | $\{0, 1e-5, 1e-4, 1e-3\}$ | 1e-5 | 1e-5 | 1e-4 |

## C.3.4   AGAC

We identified the entropy coefficient, intrinsic coefficient and learning rate of the predictor ('adversary' in the original terminology) through a grid search.

AGAC also introduces a hyperparameter called the 'episodic count coefficient' to scale the bonus from the episodic counts term.

Our main results in Figure 2 are obtained without the use of episodic counts (this coefficient is set to zero). To report results with episodic counts in Figure 3 we search over values of this hyperparameter.

Tables 12 and 13 describe the search grid and selected hyperparameters for AGAC.

Note that in the experiments without episodic counts, AGAC did not succeed (with any search configuration) on any environment other than ObstructedMaze-2Dlh.

Table 12: AGAC specific hyperparameters for MiniGrid experiments with egocentric observations (without counts).

| Hyperparameter | Candidates | Without episodic counts | | |
| --- | --- | --- | --- | --- |
| | | MultiRoom | KeyCorridor | ObstructedMaze |
| Predictor learning rate $\eta_p$ | $\{0.25\eta, 0.5\eta\}$ | $0.25\eta$ | $0.25\eta$ | $0.25\eta$ |
| Intrinsic coefficient | $\{0.00001, 0.00005, 0.0001, 0.0005\}$ | 0.00005 | 0.00005 | 0.00005 |
| Episodic count coefficient | $\{0\}$ | 0 | 0 | 0 |
| Entropy coefficient | $\{0, 1e-5, 1e-4, 1e-3\}$ | 1e-5 | 1e-4 | 1e-4 |

## C.4   Grid search and selected hyperparameters for analysis and ablation

For the analysis of discount factors presented in Figures 4a and 4b we conducted a search over best intrinsic reward coefficients ($\beta$) for each GVF discount factor $\gamma_z$. This was necessary as the magnitudes of the TD-error are sensitive to the choice of $\gamma_z$. The candidates and selected intrinsic reward coefficients ($\beta$) are reported in Table 14. The remaining hyperparameters are the same as the

Table 13: AGAC specific hyperparameters for MiniGrid experiments with episodic counts

| Hyperparameter | Candidates | With episodic counts | | |
| --- | --- | --- | --- | --- |
| | | MultiRoom | KeyCorridor | ObstructedMaze |
| Predictor learning rate $\eta_p$ | $\{0.25\eta, 0.5\eta\}$ | $0.25\eta$ | $0.25\eta$ | $0.25\eta$ |
| Intrinsic coefficient | $\{0.00001, 0.00005, 0.0001, 0.0005\}$ | 0.00005 | 0.00005 | 0.00005 |
| Episodic count coefficient | $\{0.0005, 0.001, 0.005, 0.01\}$ | 0.005, | 0.001 | 0.001 |
| Entropy coefficient | $\{0, 1e-5, 1e-4, 1e-3\}$ | 1e-5 | 1e-4 | 1e-4 |

ones selected for RC-GVF with panoramic views (see Table 7). Final results are reported with 10 independent runs of the selected hyperparameters.

Table 14: Search over intrinsic coefficients for different GVF discount factors. KCS5R3 refers to the KeyCorridor-S5R3 environment and MRN12S10 refers to the MultiRoom-N12-S10 environment.

| GVF $\gamma_z$ | Intrinsic coefficients ($\beta$) considered | Selected $\beta$ (KCS5R3) | Selected $\beta$ (MRN12S10) |
| --- | --- | --- | --- |
| 0.99 | { 1e-6, 5e-6, 1e-5, 5e-5, 1e-4, 5e-4, 1e-3, 5e-3} | 1e-5 | 1e-4 |
| 0.95 | $\{0.0005, 0.001, 0.005, 0.01, 0.05\}$ | 0.001 | 0.001 |
| 0.7 | $\{0.1, 0.5, 1\}$ | 0.5 | 0.1 |
| 0.6 | $\{0.1, 0.5, 1\}$ | 1 | 0.1 |
| 0.3 | $\{0.5, 1, 5, 25, 50\}$ | 10 | 1 |
| 0.1 | $\{1, 5, 10, 25, 50, 100\}$ | 25 | 5 |
| 0.05 | $\{1, 5, 10, 25, 50, 100\}$ | 25 | 10 |
| 0.0 | $\{5, 10, 25, 50, 100\}$ | 50 | 10 |

We conduct a similar search over choices of intrinsic coefficients for the ablation with components of RC-GVF (see Figure 5). The candidates and selected intrinsic reward coefficients are reported in Table 15. The hyperparameters for RC-GVF ($\gamma_z = 0.6$) and RND are the same as the ones reported in Tables 7 and 9 respectively. Final results are reported with 10 independent runs of the selected hyperparameters.

Table 15: Search over intrinsic coefficients for different variants of RC-GVF. KCS5R3 refers to the KeyCorridor-S5R3 environment.

| Approach | Intrinsic coefficients ($\beta$) considered | Selected $\beta$ (KCS5R3) |
| --- | --- | --- |
| RC-GVF ($\gamma_z = 0.0$) | $\{5, 10, 25, 50, 100\}$ | 50 |
| RC-GVF ($\gamma_z = 0.0$) w/o disagreement | $\{0.0005, 0.001, 0.005, 0.01\}$ | 0.005 |
| RC-GVF ($\gamma_z = 0.0$) w/o recurrence | $\{10, 25, 50, 100\}$ | 50 |
| RC-GVF ($\gamma_z = 0.6$) w/o disagreement | $\{0.00001, 0.00005, 0.0001, 0.0005\}$ | 0.00005 |
| RC-GVF ($\gamma_z = 0.6$) w/o recurrence | $\{0.1, 0.5, 1\}$ | 0.5 |

## C.5   Computational resources

All our experiments were conducted on a cluster with NVIDIA Pascal P100 GPUs. Due to low GPU usage of a single run, we could accommodate multiple concurrent runs (usually 5) on the same node (CPU + GPU). Our final experiments should be easy to replicate even on a single GPU machine. A single run of RC-GVF on a desktop machine with an NVIDIA GeForce RTX 2080 GPU completes about 30 million frames in MiniGrid (with egocentric observations) in about 9 hours (equivalent to our KeyCorridor experiments).

## D   Additional experiments with the diabolical lock problem

### D.1   Further baselines

Our aim with the diabolical lock experiment is to illustrate the differences between RND and RC-GVF. For completeness, in this section, we additionally report the performance of AGAC and NovelD. We

also investigate the performance of RND with a higher capacity predictor, as used by RC-GVF (see Appendix B).

We conduct a grid-search to select the values for hyperparameters specific to the new baselines. These are described in Tables 17, 18, and 19. The best configuration is selected by evaluating average lock completion for each candidate in the grid across 5 seeds. The remaining hyperparameters are the same as the ones detailed in Appendix B.2.

The farthest completions at different points in training (with 10 independent runs of the selected hyperparameters) are reported in Table 20. We observe that neither AGAC nor NovelD manage to complete the lock. AGAC does not make much progress towards solving the problem. NovelD fares better and performs roughly as well as RND. We also observe that the higher capacity predictor does not improve RND (on average).

Table 16: Mean over 10 seeds and (min, max) of the farthest column reached (of $H = 100$) at different points on the diabolical lock problem.

| Frames | 5M | 10M | 15M | 20M |
|---|---|---|---|---|
| RC-GVF | 61 (47, 81) | 94 (83, 100) | 100 (100, 100) | 100 (100, 100) |
| RND | 28 (21, 41) | 37 (27, 47) | 41 (30, 53) | 46 (33, 58) |
| RND (with high capacity predictor) | 22 (8, 41) | 28 (8, 62) | 32 (8, 70) | 34 (8, 72) |
| NovelD | 28 (22, 35) | 35 (26, 50) | 41 (27, 54) | 45 (32, 60) |
| AGAC | 5 (5, 7) | 6 (5, 7) | 6 (5, 9) | 6 (5, 9) |

Table 17: RND (with high capacity predictor) hyperparameters for diabolical lock experiments.

| Hyperparameter | Candidates | Selected |
|---|---|---|
| Predictor learning rate $\eta_p$ | $\{\eta, 0.75\eta, 0.5\eta, 0.25\eta\}$ | $0.25\eta$ ($\eta = 0.0005$) |
| Intrinsic reward coefficient $\beta$ | $\{4.0, 2.0, 1.0, 0.5, 0.25\}$ | 1.0 |

Table 18: NovelD specific hyperparameters for diabolical lock experiments.

| Hyperparameter | Candidates | Selected |
|---|---|---|
| Predictor learning rate $\eta_p$ | $\{\eta, 0.75\eta, 0.5\eta, 0.25\eta\}$ | $0.25\eta$ ($\eta = 0.0005$) |
| Intrinsic reward coefficient $\beta$ | $\{4.0, 2.0, 1.0, 0.5, 0.25\}$ | 1.0 |
| Scaling factor $\alpha$ | $\{0.5\}$ | 0.5 |

Table 19: AGAC specific hyperparameters for diabolical lock experiments.

| Hyperparameter | Candidates | Selected |
|---|---|---|
| Predictor learning rate $\eta_p$ | $\{0.5\eta, 0.25\eta\}$ | $0.25\eta$ ($\eta = 0.0005$) |
| Intrinsic reward coefficient $\beta$ | $\{4.0, 2.0, 1.0, 0.5, 0.25\}$ | 1.0 |
| Shared architecture for actor and critic | $\{True, False\}$ | $True$ |

## D.2 Ensemble size analysis

In this section, we analyse the effect of increasing the ensemble size in RC-GVF on the performance in the diabolical lock problem. We consider 2, 4, 6 and 8 predictors in the ensemble. The other hyperparameters are the same as those reported in Table 4.

The results are reported across 10 seeds in Table 20. For the most part, we obtain similar results for the different ensemble sizes. The minor exception is with 6 predictors, where one run out of the ten fails to reach the end of the lock.

Table 20: Mean over 10 seeds and (min, max) of the farthest column reached (of $H = 100$) at different points during training on the diabolical lock problem. Higher is better.

| Number of predictors / Frame | 5M | 10M | 15M | 20M |
|---|---|---|---|---|
| 2 | 61 (47, 81) | 94 (83, 100) | 100 (100, 100) | 100 (100, 100) |
| 4 | 66 (62, 78) | 96 (82, 100) | 100 (100, 100) | 100 (100, 100) |
| 6 | 61 (39, 75) | 90 (59, 100) | 96 (63, 100) | 97 (67, 100) |
| 8 | 69 (57, 84) | 94 (77, 100) | 100 (99, 100) | 100 (100, 100) |

# E  Additional experiments in MiniGrid

## E.1  Intrinsic reward from solely episodic counts

Figure 7 shows that intrinsic rewards from episodic state counts are sufficient for solving the MultiRoom-N7S8 and MultiRoom-N12S10 environments.

The exact form of the intrinsic bonus is $R_i(s_t) = \frac{\beta}{\sqrt{N_e(s_t)}}$, where $N_e(s_t)$ is the number of times the agent has been in that position in that episode, and $\beta$ is the intrinsic reward coefficient.

We perform a search over the intrinsic reward coefficients $\beta \in \{0.0005, 0.001, 0.005, 0.01\}$ and select $\beta = 0.005$ as a suitable value. The entropy coefficient for this experiment is set to 0.00001.

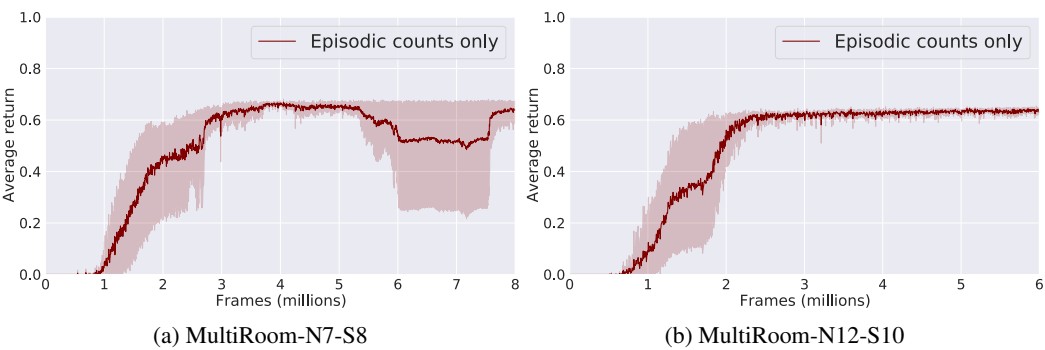

(a) MultiRoom-N7-S8         (b) MultiRoom-N12-S10

Figure 7: Performance with an intrinsic reward derived from episodic counts. These results show that a bonus term from solely episodic counts can guide useful exploration in MiniGrid environments. 95% bootstrapped confidence intervals are shown for 5 seeds.

## E.2  RC-GVF with episodic counts

Several recent works incorporate episodic counts from the simulator as part of their intrinsic reward in MiniGrid [46, 18, 77]. Our main experiments with RC-GVF do not utilise episodic counts as we do not wish to provide agents with access to privileged information. For completeness, this section reports the result from a preliminary experiment, where we study the performance of RC-GVF with the use of episodic counts. We incorporate episodic counts in the same way as for the other baselines (see Appendix C.2).

To account for change in scale of the intrinsic reward with the use of the counts, we perform a search over the intrinsic reward coefficients $\beta \in \{1, 5, 10\}$ and select $\beta = 5$ for KeyCorridor-S5R3 and $\beta = 1$ for MultiRoom-N12-S10. All other hyperparameters are the same as those reported in Table 6.

As shown in Figure 8, RC-GVF can solve the harder environments (KeyCorridor-S5R3 and MultiRoom-N12-S10) without panoramic observations when including episodic counts. In KeyCorridor-S5R3 we observe that RC-GVF adapted in this way is competitive with the baselines, while in MultiRoom-N12-S10 it converges more slowly. This is unsurprising since we have not attempted to make our approach amenable to incorporating episodic counts. Indeed, in MultiRoom-N12-S10, we observe that RC-GVF performs worse compared to when only using episodic state-counts (and no value function prediction), indicating that some interference takes place between these two signals. Thus, we expect that other ways of combining (or weighting) these signals (such

as by using an additive term for the state-counts as opposed to a multiplicative one) should lead to improved performance when also including state-counts.

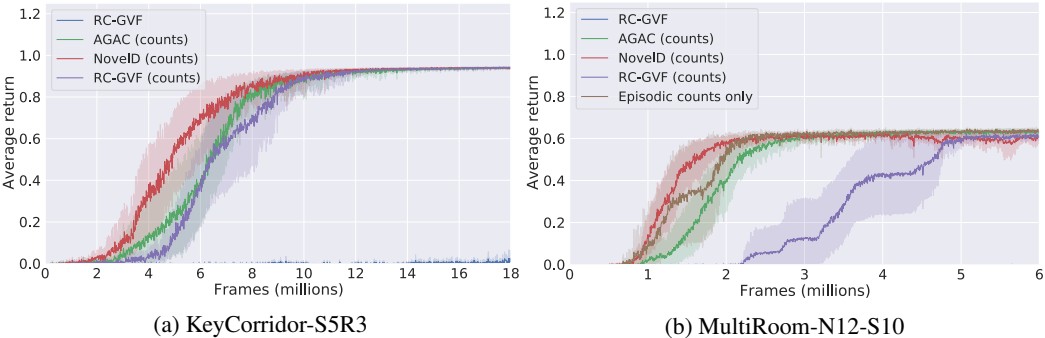

(a) KeyCorridor-S5R3          (b) MultiRoom-N12-S10

Figure 8: These results show that the performance of RC-GVF can also be improved through the use of episodic counts. All approaches use the standard egocentric observations. 95% bootstrapped confidence intervals are shown for 10 seeds.

### E.3  Improvement from history conditioning

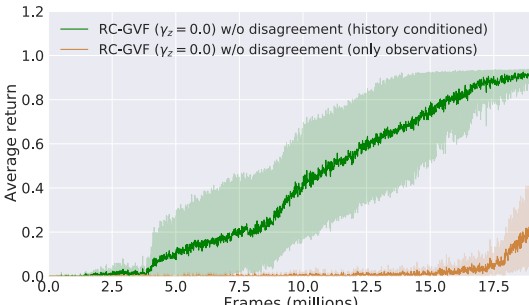

Figure 9: Studying the performance of RC-GVF ($\gamma_z = 0$) without the disagreement term on the KeyCorridor-S5R3 environment. We see that our recurrent predictor conditioned on histories of observations, actions and pseudo-rewards performs better than one conditioned on just histories of observations. 95% bootstrapped confidence intervals are shown for 10 seeds.

As indicated in Figure 5a, the recurrent predictor plays an important role in the functioning of RC-GVF. Here we present a result which shows that a recurrent predictor conditioned on histories of observations, actions and pseudo-rewards outperforms a recurrent predictor operating on histories of observations. Figure 9 compares the performance of the two variants on the KeyCorridor-S5R3 environment. We hypothesise that the improved results with history conditioning arise from previous pseudo-rewards being made available to the predictor. In a procedurally generated environment, having access to previous pseudo-rewards could support predictions of the new input observations on which the predictor has not yet been trained.

### E.4  Ensemble size analysis

Figure 10 shows the performance of larger ensemble sizes with RC-GVF in two MiniGrid environments (KeyCorridor-S5R3 and MultiRoom-N12-S10). We observe similar results for ensemble sizes of 2, 4, 6 and 8 predictors. The intrinsic coefficient $\beta = 1$ in KeyCorridor-S5R3 and $\beta = 0.5$ in MultiRoom-N12-S10. The remaining hyperparameters are the same as the ones in Table 7.

These results indicate that two member ensembles usually suffice in these problems. We speculate how measuring disagreement with an ensemble of two predictors might be sufficient when also including the prediction error term. However, note that the ensemble members have a common recurrent core and only differ in MLP heads; perhaps having separate RNNs for each member would lead to greater variation with ensemble size.

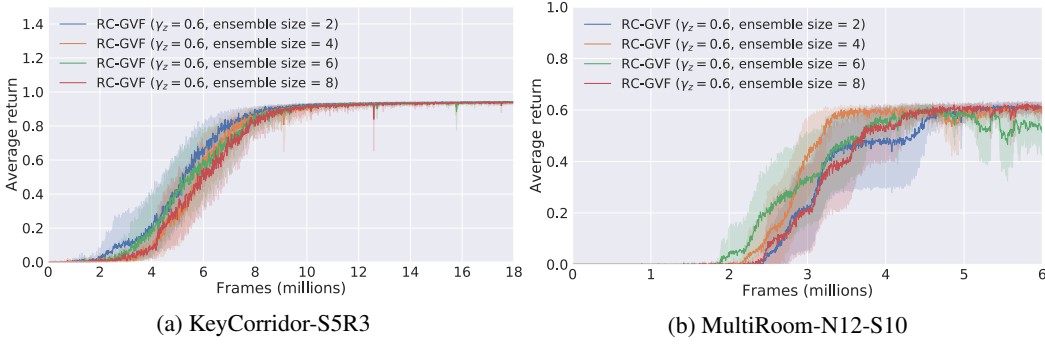

(a) KeyCorridor-S5R3

(b) MultiRoom-N12-S10

Figure 10: Comparison of RC-GVF with different numbers of predictors in the ensemble. 95% bootstrapped confidence intervals are shown for 10 seeds.