# OpenReview forum: "Exploring through Random Curiosity with General Value Functions"
_NeurIPS.cc/2022/Conference — NeurIPS 2022 Accept_

### Official Review · Reviewer_5dYH · 2022-07-07

**Rating:** 6
**Confidence:** 3
**Soundness:** 2 fair
**Presentation:** 3 good
**Contribution:** 2 fair

**Summary:**

The authors propose a novel form of artificial curiosity that utilizes randomly initialized neural networks to represent generalized value functions. The authors utilize the insight that temporally extended predictions of environment quantities (such as state transitions, etc.) are often useful for generating artificial curiosity in partially observable environments. However, instead of using state sequences directly, they utilize randomly initialized NNs to annotate trajectories with pseudo-rewards and utilize the uncertainty in the prediction of these pseudo-rewards by a prediction network ensemble as a signal for artificial curiosity. They evaluate their method in lock-opening environments and in grid world tasks and show a performance improvement on curiosity baselines.

**Questions:**

1. Are there any ablations on ensemble size?
2. As mentioned I would be interested in knowing the results of AGAC and NovelD on the lock task.


**Limitations:**

I believe the limitations of the method have been adequately discussed by the authors. I believe the paper is well written. While not state-of-the-art in terms of results, I believe there is some novelty in the method. I am on the fence about my decision about the paper.

**Strengths And Weaknesses:**

Strengths:
1. The paper is well motivated. Long-term dependencies should be accounted for when generating artificial curiosity signals.
2. The writing is easy to follow and the approach is intuitive.
3. The experiments are representative of hard exploration problems.
Weaknesses:
My only real qualm with the paper is the significance of the results. The method strongly outperforms the baseline in the lock task but really only outperforms the baselines in 2 of the grid-world tasks. Additionally, why are there no performance metrics for AGAC and NovelD on the lock task? I would be interested in knowing those numbers as a reader.

---

> ### Author Response · Authors · 2022-08-02
> **Individual response to Reviewer 5dYH**
>
> Thank you for your review and valuable feedback. We are glad that you found our paper novel, well-motivated, and easy to follow. We understand that you are on the fence about our work and we hope that our response and revision may help you to decide to endorse our submission.
>
> In your review, you mention that **“My only real qualm with the paper is the significance of the results.  The method strongly outperforms the baseline in the lock task but really only outperforms the baselines in 2 of the grid-world tasks.”**
>
> Indeed, RC-GVF does not solve all environments in the Minigrid suite. However, as is evident from experiments, state of the art methods such as NovelD and AGAC also cannot solve these environments (without relying on episodic state counts), and performs worse in those environments that RC-GVF solves. Based on your suggestion, we have now also compared RC-GVF to NovelD and AGAC on the diabolical lock environment where we similarly find that RC-GVF is the best performing method.
>
> Other than proposing a novel approach to generating intrinsic rewards that extends RND to using random GVFs, we wanted to push the development of future exploration methods to consider the more natural setting without episodic state counts. With RC-GVF achieving SOTA in this setting we contribute a useful baseline for others to improve on, while also pointing to deficits in existing methods applied to this setting.
>
> We provide responses to your specific questions below.
>
> **“Are there any ablations on ensemble size?”**
>
> Thank you for suggesting this analysis. We performed a study with ensemble sizes 4, 6 and 8 on KeyCorridor-S5R3 and MultiRoom-N7S8 and observed negligible benefits with higher ensemble sizes (KeyCorridor-S5R3: https://ibb.co/X5bKCYb and MultiRoom-N12S10: https://ibb.co/L8qbDH9). This appears to be in line with recent work that found a two member ensemble sufficient for balancing optimism and pessimism with distributional estimates of the return [Moskovitz 2021].
>
> We also conducted an analysis on the lock task, where present results for lock completion at 15M frames and we found small improvements with larger ensembles of 4 and 6 predictors.
>
> | # predictors | Mean and (min, max) of farthest column | Runs that reach end of the lock |
> | --------------- | ---------------------------------------------------- | ----------------------------------------- |
> |         2          |   		94 (66, 100)		      |			6/10		|
> |          4	           | 			98 (90,100)		     |			8/10		|
> |          6 	|		95 (65, 100)		    |  			8/10		|
>
> **“As mentioned I would be interested in knowing the results of AGAC and NovelD on the lock task.”**
>
> Thank you for suggesting this. Our initial aim with the diabolical lock experiment was to illustrate differences between RND and RC-GVF, since RND is often evaluated in this environment, while approaches like AGAC and NovelD are not.
>
> However, we agree it is valuable to have some indication of how they perform on the lock task. We present the results for lock completion at 15M frames for these approaches and RC-GVF in the table below.
>
> | Model | Mean and (min, max) of farthest column | Runs that reach end of the lock |
> | --------------- | ---------------------------------------------------- | ----------------------------------------- |
> | RC-GVF      |   		94 (66, 100)		      |			6/10		|
> |          ACAG           | 		5 (4, 6)		     |			0/5	|
> |          NovelD 	|		24 (22, 31)		    |  			0/5		|
>
> As we can see, AGAC does not make progress towards solving the problem (for a wide range of hyperparameters). NovelD fares better, solving about 30% of the lock on the best run. Here we explored different hyperparameters ranges for the intrinsic coefficient and the learning rate of the predictor. For NovelD we used the suggested default of 0.5 for the scaling coefficient (\alpha). These are results with 5 seeds, we will include final results with 10 seeds in the supplementary material.

---

> > ### Comment · Reviewer_5dYH · 2022-08-09
> > **Thanks for your rebuttal**
> >
> > Thank you for your response. I believe the idea is novel and given the experimental results, I am happy to increase my score.

---

### Official Review · Reviewer_9UDz · 2022-07-10

**Rating:** 7
**Confidence:** 4
**Soundness:** 3 good
**Presentation:** 3 good
**Contribution:** 4 excellent

**Summary:**

This paper proposes a generalisation of the random network distillation (RND) to deal with partially-observable environments. That is, the paper extends the notion of intrinsic rewards in RND to deal with long-horizon, sparse reward environments that are partially observable by (1) predicting the value of pseudo-rewards (as opposed to per-state rewards in RND), (2) using a recurrent value predictor network to create a belief state due to partial observability, and (3) using the disagreement among an ensemble of value predictors to explore where predictions are epistemically uncertain. The combination of this approach with any model-free on-policy RL agent as a basis leads to an approach to exploration through temporally-extended curiosity. To validate the approach, the authors have combined their approach with PPO and tested it on a long-horizon anti-reward-shaped task, as well as on a set of MiniGrid tasks with egocentric and panoramic observations. They conduct analysis and ablation studies to verify that all added components are necessary to achieve the best performance in such tasks over RND and other techniques.

**Questions:**

1. In order to avoid state aliasing in the rewards, wouldn't it make more sense for the rewards to also be generated through a fixed, randomly initialised recurrent network that conditions on the sequence of observations, actions, and pseudo-rewards? For example, say there are only aliasing states between a trajectory that the agent has experienced repeatedly and a trajectory that the agent had not experienced before. Now, the history summaries would be the same and so would the value predictions, and the disagreement would be low. But if the trajectories were distinguishable by different pseudo-rewards, then the histories would be distinct and the epistemic uncertainty would be high for the trajectory that had not been explored before.

2. The authors have tried different discount factors for training the GVF predictors and argue that lower discounts are not good because the prediction horizon should be larger. Couldn't the worse performance of lower discount factors be due to optimisation issues under function approximation? (See van Seijen et al. (2019) for details on this issue.)

3. I also feel the range of discount factors used is somewhat arbitrary and not well motivated. The most commonly-used discount factors (e.g. $\gamma = 0.9$ or $\gamma = 0.99$) are missing in the tested values in Figure 4. Why? I understand that these are not agent discounts and are only used for exploration, but I cannot process why still a higher value shouldn't in principle be preferred in episodic settings. I think too high wouldn't work well with function approximation due to optimisation issues, but a value of 0.99 should work in practice. Why is it not preferred in principle by the authors?

4. Does the basis PPO agent use the history summary or only use the latest observation? Is this setting the same among all baselines?

5. Where can I find the result that you refer to in footnote 1?

6. The approach seems also applicable in principle to continuous state-action environments. The authors do not claim this, but could you comment on whether you see any obstacles in such environments?

7. How do you justify using only 10 independent trials (or only 5 in one task)? PPO itself generally has very high variance across trials, and my experience is that using 20 trials is important in deriving any conclusions about performance characteristics. Now, in your context, the variation added due to pseudo-reward predictors could only make this worse. It is my view that, given that the domains are not too computationally expensive, it is critical to at least make sure the results are statistically reliable. Please share your view on this, and why the choice to run only 10/5 seeds.

Should these questions be adequately clarified/addressed, I'd be happy to increase my score.

- H. van Seijen, M. Fatemi, A. Tavakoli. “Using a Logarithmic Mapping to Enable Lower Discount Factors in Reinforcement Learning.” NeurIPS (2019).

**Limitations:**

The authors discuss some limitations of their work. I have asked for comments on other potential limitations in the Questions section.
They appropriately state that the nature of the work does not per se entail a potential negative societal impact.

**Strengths And Weaknesses:**

### Strengths:

- The paper moves in the direction of extending a simple yet powerful idea (namely, RND) to the more general problem setting of partially-observable environments with sparse rewards.

- The approach is simple and fundamentally well justified; specifically, (1) GVF formulation is natural for going beyond predicting single-step rewards (as in RND), (2) using random targets as in RND has been previously shown effective (and simplifies the problem of selecting/discovering auxiliary tasks in Horde-style architectures), (3) using recurrence as agent's state-update function to deal with partial-observability is well-accepted and the authors have also justified its particular importance in the context of this paper, (4) using prediction disagreement among an ensemble is an effective proxy for epistemic uncertainty.

- The approach is well-motivated through illustrative examples and challenging exploration problems (but with relatively simplistic dynamics; namely, Diabolical Locks and 6 MiniGrid environments).


### Weaknesses:

- I feel that the pseudo-reward generator should also be based on some representation of history (e.g. via a fixed and randomly initialised recurrent network). I discuss this further in the Questions section (first question).

- The trade-offs involved in choosing the discount factor are not discussed to a good extent. For example, why should we even care for low discounts? Why do they think $\gamma = 0.6$ works better, and not $\gamma = 0.9$ or $\gamma = 0.99$? Discussing intuitions and drawing on previous literature on the role of discounting (especially in combination with function approximation) could help improve this shortcoming.

- In relation to discounting, I also feel the range of discount factors used is somewhat arbitrary and not well motivated. The most commonly-used discount factors are missing in the tested values in Figure 4. Why?

- I feel that the panoramic-view experiment or at least the justifications around it are forced. The statement in footnote 1 of the paper is not a reasonable argument in my view. If the environment doesn't give you some information, assuming anything beyond it is privileged. Augmenting the agent with more sensors to get a panoramic view is privileged too. As such, I much rather see the result of your approach having access to the episodic count as opposed to a panoramic view.

---

> ### Author Response · Authors · 2022-08-02
> **Individual response to Reviewer 9UDz (1/3)**
>
> Thank you very much for your detailed review and intriguing questions. We are glad that you feel positively about our paper and that you found our approach interesting and well-justified.
>
> We discuss the recurrent pseudo-reward generator and new results regarding the GVF discount factor (and motivation) in the response to your questions down below.
>
> Regarding your comments on the experiments with panoramic observations, we agree that this setting is not ideal and perhaps somewhat forced. There are two matters at hand here:
>
> * Somehow prior approaches to exploration have incorporated privileged information about episodic state counts as part of their approach, which greatly simplifies the exploration problem in a way that is likely difficult to achieve in the real world. Indeed, in Figure 6 in appendix B we present clear evidence that often these counts alone already lead to a very successful intrinsic reward. Importantly, without using these episodic counts we observe that existing approaches struggle a lot, while RC-GVF performs very well (Figure 2).
> * The second issue (2) we encountered, which was also encountered in prior work, is that rotations in MiniGrid affect the observation in a detrimental way, which is certainly not representative of how real-world agents (including robots) sense the world. This led us to adopt the panoramic setting introduced in prior work [38]. While we acknowledge that this can also be viewed as using some sort of privileged information, we would also argue that this assumption is a lot more reasonable. In part because it effectively translates to equipping the agent with additional sensors, and because it may be achieved with representation learning techniques that consider symmetries like rotation invariance.
>
> We wanted to avoid future work comparing to RC-GVF using episodic counts, which is why we were hesitant to report these numbers in the paper on KeyCorridor. We have now added these results as requested and discuss them in reply to your other comment down below.
>
> **“In order to avoid state aliasing in the rewards, wouldn't it make more sense for the rewards to also be generated through a fixed, randomly initialised recurrent network that conditions on the sequence of observations, actions, and pseudo-rewards? For example, say there are only aliasing states between a trajectory that the agent has experienced repeatedly and a trajectory that the agent had not experienced before. Now, the history summaries would be the same and so would the value predictions, and the disagreement would be low. But if the trajectories were distinguishable by different pseudo-rewards, then the histories would be distinct and the epistemic uncertainty would be high for the trajectory that had not been explored before.”**
>
> Using a randomly initialized RNN to generate pseudo-rewards is an interesting idea that we have not considered. While there may be situations where recurrent pseudo-rewards are beneficial, we could not think of such a scenario (once the predictor is conditioned on histories).
>
> We are not fully sure that we understood the details of the example that you presented. Perhaps you could clarify the situation where history summaries would be the same, but the trajectories can/should be distinguishable?
>
> As you mention, using a stateless pseudo-reward generator could lead to the same pseudo-reward for different states emitting the same observation, however, it is in our view unlikely that the longer term pseudo-return following from the observation would also be aliased.

---

> > ### Author Response · Authors · 2022-08-02
> > **Individual response to Reviewer 9UDz (2/3)**
> >
> > **“The authors have tried different discount factors for training the GVF predictors and argue that lower discounts are not good because the prediction horizon should be larger. Couldn't the worse performance of lower discount factors be due to optimisation issues under function approximation? (See van Seijen et al. (2019) for details on this issue.)”**
> >
> > Thank you for pointing us to this reference. It is an interesting aspect that could impact results. We believe that some of these factors might be less influential in our GVF prediction case, unlike analysis in the control setting by van Seijen et al. (2019). In our prediction case, there would not be effects from the mismatch between the finite horizon undiscounted return (used for measuring performance) and the discounted return. Similarly, optimization effects arising from action gaps might be more relevant in the learning of optimal value functions. However, there could also be optimization effects even in the GVF prediction case with function approximation. For example, there could be scenarios where different behaviors or policies are not reflected in the value function with certain discount factors. These effects could certainly be interesting to consider in future work and we will comment on this in the paper.
> >
> > More generally, as we mention in the limitations section, moving away from single fixed discount factors seems like a good idea. Some possibilities include adapting discount factors during learning, considering GVFs under multiple discount factors, or utilizing state dependent discounts.
> >
> > **“I also feel the range of discount factors used is somewhat arbitrary and not well motivated. The most commonly-used discount factors (e.g. γ = 0.9 or γ = 0.99) are missing in the tested values in Figure 4. Why? I understand that these are not agent discounts and are only used for exploration, but I cannot process why still a higher value shouldn't in principle be preferred in episodic settings. I think too high wouldn't work well with function approximation due to optimisation issues, but a value of 0.99 should work in practice. Why is it not preferred in principle by the authors?”**
> >
> > Thank you for pointing this out. We agree that further studying the effects of the discount factor is interesting and useful. We now present results with higher discount factors (0.99 and 0.95) in our discount analysis with KeyCorridorS5R3 and MultiRoom-N7S8. The new figures are available at https://ibb.co/SmWsNLH (KeyCorridor) and https://ibb.co/bHBQDXR (MultiRoom). 0.95 does quite well in KeyCorridor but not in the MultiRoom. 0.99 is worse than values like 0.6/0.7 in both environments.
> >
> > One reason could be that while predicting more than one step into the future is useful, predicting too far ahead into the future makes the task too difficult, especially in procedurally generated environments like MiniGrid.
> >
> > **“Does the basis PPO agent use the history summary or only use the latest observation? Is this setting the same among all baselines?”**
> >
> > The base PPO agent is always a recurrent network operating on the history of observations in the MiniGrid experiments. This holds true for all baselines as well.
> >
> > **“Where can I find the result that you refer to in footnote 1?”**
> >
> > This statement was based on preliminary experiments on KeyCorridorS5R3 that we had not included in the initial submission. We now provide results for RC-GVF with episodic counts and egocentric observations (based on a coarse grid search for intrinsic coefficients) on two of the harder environments (KeyCorridor-S5R3 and MultiRoom-N12S10).
> >
> > On KeyCorridor (https://ibb.co/Gx0PJrN), we note how RC-GVF achieves competitive performance, although performs slightly worse compared to when using panoramic observations. We also ran an additional experiment using episodic state counts on MultiRoom-N12-S10 (https://ibb.co/VCrcCcv). We observe that the performance is worse compared to when only using episodic state-counts, indicating that some interference takes place between these two signals. Indeed we have not attempted to make our approach amenable to incorporating episodic counts previously: we heuristically applied a scaling term similar to the one described by [Raileanu et al., 2020], where the intrinsic reward is divided by the square root of the episodic count. Given the baseline performance when only using counts it is reasonable to expect that other ways of combining (or weighting) these components (eg. using an additive term for the state-counts as opposed to a multiplicative one) can lead to substantially improved performance in this environment.

---

> > > ### Author Response · Authors · 2022-08-02
> > > **Individual response to Reviewer 9UDz (3/3)**
> > >
> > > **“The approach seems also applicable in principle to continuous state-action environments. The authors do not claim this, but could you comment on whether you see any obstacles in such environments?”**
> > >
> > > Thank you for pointing this out. We have not tested RC-GVF in continuous environments (although the diabolical lock has smooth noisy observations). However, since we focus on prediction problems (GVFs), this approach could almost immediately apply to continuous control. One aspect to consider is that if we wanted to include control GVFs to learn optimal value functions for certain cumulants, we would have to make modifications to deal with maximization over continuous actions. Nonetheless this is an interesting direction for future work that we will comment on in the paper.
> > >
> > > **“How do you justify using only 10 independent trials (or only 5 in one task)? PPO itself generally has very high variance across trials, and my experience is that using 20 trials is important in deriving any conclusions about performance characteristics. Now, in your context, the variation added due to pseudo-reward predictors could only make this worse. It is my view that, given that the domains are not too computationally expensive, it is critical to at least make sure the results are statistically reliable. Please share your view on this, and why the choice to run only 10/5 seeds.”**
> > >
> > > In principle we fully agree with this sentiment as there are indeed variance and stability issues with PPO. Our current choice of 10 seeds (for all experiments apart from OM2Dlhb) is mainly a pragmatic one (to reduce the computational cost), yet it is already nearly twice as many as is typically encountered in relevant prior works where 4 to 6 seeds are the default [Zhang et al., 2021; Flet-Berliac et al., 2021; Raileanu et al., 2020]. One way in which we have further attempted to mitigate variance and bias in report results is by reporting all results with fresh runs after the initial hyperparameter selection.
> > >
> > > While we are eager to provide 20 definitive runs wherever possible for a final version, we would like to note that, although the Minigrid environments are simple in appearance, they pose challenging exploration problems that require many million frames of training.
> > >
> > > Zhang, T., Xu, H., Wang, X., Wu, Y., Keutzer, K., Gonzalez, J. E., & Tian, Y. (2021). Noveld: A simple yet effective exploration criterion.
> > >
> > > Raileanu, R., & Rocktäschel, T. (2020). Ride: Rewarding impact-driven exploration for procedurally-generated environments.
> > >
> > > Flet-Berliac, Y., Ferret, J., Pietquin, O., Preux, P., & Geist, M. (2021). Adversarially guided actor-critic.

---

> > > > ### Comment · Reviewer_9UDz · 2022-08-03
> > > > **Thanks for your response!**
> > > >
> > > > Thanks for your detailed response and the additional results. Your response has adequately addressed my main concerns. While I do appreciate the links to new results (very convenient), I also would've really liked to see the suggested changes and new results reflected in an updated PDF. Nevertheless, I do trust that the authors would make appropriate changes in the final version and as such I'm happy to increase my score at this point.

---

### Official Review · Reviewer_ZFPx · 2022-07-11

**Rating:** 4
**Confidence:** 3
**Soundness:** 2 fair
**Presentation:** 2 fair
**Contribution:** 2 fair

**Summary:**

The authors propose Random Curiosity with General Value Functions (RC-GVF) that essentially extends the intrinsic rewards obtained through Random Network Distillation (RND) to temporally extended outcomes of action sequences, in order to tackle long-horizon hard-exploration tasks in partially observable environments. To deal with aleatoric uncertainty, an ensemble of predictors is trained and the ensemble disagreement is combined with the prediction error as a multiplicative factor. The method is evaluated in two sets of environments: the diabolical lock and the MiniGrid environments.

**Questions:**

- Why is there no recurrency in the predictor network for the diabolical lock experiment?
- Why are there only two ensemble members? How does the number of ensemble members affect performance in regards to noise levels in the tested environments?
- I find the ablation study Figure 4c to be contradicting the statements made in section 3.3. Using $\gamma_z=0$ is the equivalent of intrinsic rewards by RND, in which case as the authors state no ensemble is needed (The deterministic target value of the pseudo-reward takes care of the stochasticity). So why is the ensemble ablation performed for this discount value? In fact why are there no ablations for $\gamma_z=0.6$?
- If I understand it correctly, the main difference between RC-GVF with $\gamma_z=0$ with and without recurrence is that the predictor network unlike in RND is not the same as the pseudo-reward generator and in the case of Minigrid doesn’t use an LSTM core. Did you do an ablation on the extra representational power in the different predictor network architecture of RC-GVF compared to the pseudo-reward generator also for the diabolical lock experiment?

**Limitations:**

-

**Strengths And Weaknesses:**

I believe this is an interesting work touching on an important subject that is utilizing future novelty with long horizon novelty predictions to solve hard-exploration tasks.

However, I believe that the authors of the paper didn’t discuss the related work in this direction in enough detail. There are several recent papers that build upon the ensemble disagreement proposed in [1] (which is discussed in the current paper) and extend it to get multi-step novelty estimates into the future, such as [2] and [3]. (Especially in [3], multi-step novelty with RND as intrinsic reward is also proposed.). I believe these works should also be part of the discussion to give a more complete picture of long-horizon intrinsically-motivated exploration.

I still believe that the way the authors combined the RND intrinsic reward with general value functions is interesting and different in that they do not rely on model rollouts.

[1] Pathak, Deepak, Dhiraj Gandhi, and Abhinav Gupta. “Self-supervised exploration via disagreement.” International conference on machine learning. PMLR, 2019.

[2] Sekar, Ramanan, et al. “Planning to explore via self-supervised world models.” International Conference on Machine Learning. PMLR, 2020.

([3] Lambert, Nathan, et al. “The Challenges of Exploration for Offline Reinforcement Learning.” arXiv preprint arXiv:2201.11861 (2022).)

---

> ### Author Response · Authors · 2022-08-02
> **Individual response to Reviewer ZFPx (1/2)**
>
> Thank you for your review, suggestions, and questions. We are glad that you felt that the problem setting is important and that our approach is interesting and novel. We reply to your questions and concerns about missing related work below.
>
> **“I believe that the authors of the paper didn’t discuss the related work in this direction in enough detail. There are several recent papers that build upon the ensemble disagreement proposed in [1] (which is discussed in the current paper) and extend it to get multi-step novelty estimates into the future, such as [2] and [3]. (Especially in [3], multi-step novelty with RND as intrinsic reward is also proposed.). I believe these works should also be part of the discussion to give a more complete picture of long-horizon intrinsically-motivated exploration.”**
>
> Thank you for pointing out this missing related work. Indeed, these are relevant works to discuss and contrast with our own proposal for generating long-horizon intrinsically-motivated intrinsic rewards. It appears that these incorporate rollout based planning to optimize for long term novelty instead of only considering novelty on states that were actually encountered. This can be useful to overcome the reactive nature of model-free approaches and incorporate model-based planning into exploration.
>
> A crucial difference between these approaches and ours is that our state novelty signal itself incorporates long-term information without requiring step-by-step rollouts. Indeed, the very recent work by Lambert et al. [3] applies RND on imagined data in an interesting way and optimizes long-term novelty. However, this technique is very different from measuring a single observation’s novelty through multiple steps. In principle, it could also be possible to incorporate our long-term intrinsic reward from RC-GVF to be optimized by multi-step model based planning approaches. Thus combining these techniques could be an interesting direction for future work.
>
> We will update the revised paper to include these references and a discussion.
>
> **“Why is there no recurrency in the predictor network for the diabolical lock experiment?”**
>
> We mainly adopted this choice for compatibility reasons, because prior work evaluating on the diabolical lock did not use recurrence either [Misra et al., 2020]. Note also that there is no recurrence in the policy either (similar to in prior works), which is sufficient to solve this task.
>
> One possible reason for this choice by Misra et al. could be the nature of the partial observability in this problem, which follows a block MDP rather than one where states are completely aliased. More generally, we think it is possible that there could be a benefit from recurrence, especially with higher levels of noise, although we have not experimented with this.
>
> Misra, D., Henaff, M., Krishnamurthy, A., & Langford, J. (2020). Kinematic state abstraction and provably efficient rich-observation reinforcement learning
>
> **“Why are there only two ensemble members? How does the number of ensemble members affect performance in regards to noise levels in the tested environments?”**
>
> We mainly used only two ensemble members because this is sufficient to have some form of disagreement, and preliminary experiments on MiniGrid did not suggest a clear benefit when increasing the number of ensemble members further. This is also in line with recent work that found a two member ensemble sufficient for balancing optimism and pessimism with distributional estimates of the return [Moskovitz 2021].
>
> Based on your feedback, we have conducted experiments where we investigate the effect of ensemble sizes. In the KeyCorridor-S5R3 and MultiRoom-N12S10 environments we obtain comparable results for ensemble 2, 4, 6 and 8 ensemble members (KeyCorridor-S5R3: https://ibb.co/X5bKCYb and MultiRoom-N12S10: https://ibb.co/L8qbDH9).
>
> In the diabolical lock environments, we observed small improvements with increased ensemble sizes of 4 and 6. Here we present results of lock completion at 15M frames with 2 (original), 4 and 6 members in the ensemble as in Table 1:
>
> | # predictors | Mean and (min, max) of farthest column | Runs that reach end of the lock |
> | --------------- | ---------------------------------------------------- | ----------------------------------------- |
> |         2          |   		94 (66, 100)		      |			6/10		|
> |          4	           | 			98 (90,100)		     |			8/10		|
> |          6 	|		95 (65, 100)		    |  			8/10		|
>
>
> As indicated in your question, this could be related to the noisy observations in the environment.

---

> > ### Author Response · Authors · 2022-08-02
> > **Individual response to Reviewer ZFPx (2/2)**
> >
> > **“I find the ablation study Figure 4c to be contradicting the statements made in section 3.3. Using γ = 0 is the equivalent of intrinsic rewards by RND, in which case as the authors state no ensemble is needed (The deterministic target value of the pseudo-reward takes care of the stochasticity). So why is the ensemble ablation performed for this discount value? In fact why are there no ablations for γ = 0.6? “**
> >
> > Thank you for this comment and question. It is a valid point that we would like to address.
> >
> > In Section 3.3, we state that, “Note that in this case using the TD-error between the predictor and target is equivalent to intrinsic rewards provided by random network distillation (RND; Equation 2).”
> >
> > This equivalence would apply if we were using solely the TD-errors as intrinsic rewards for RC-GVF. Based on the definition of the RC-GVF intrinsic reward, we would still have the ensemble disagreement term with RC-GVF and γ = 0.
> >
> > In principle, there should be no need for the disagreement term with RND (no aleatoric uncertainty). However, our results do indicate a relatively minor benefit from the inclusion of the ensemble disagreement term in 4c. We hypothesize that these benefits could arise from using multiple predictors to estimate the first error term. Such reasoning would also be supported by the uncertainty analysis presented by Ciosek et al. (2020). From an implementation perspective, we also equip RC-GVF with a history conditioned LSTM predictor in the MiniGrid experiments.
> >
> > To summarize, there are two differences between RC-GVF with γ = 0 (in minigrid) and RND, (1) The history conditioned LSTM predictor (2) Disagreement term from the ensemble. The ablation in Figure 4c moves from RC-GVF to RND. Thus, we first presented the move to γ = 0, and then removed either the disagreement term or recurrent predictor. Removing both would simply be RND, for which we provide results as well.
> >
> > Regarding your comment about the ablations at γ = 0.6, this is a good point and something that we overlooked. We have run this experiment and obtained the following results (figure https://ibb.co/z21MtDD). Removing either the disagreement term or the recurrent predictor worsens the performance of RC-GVF with γ = 0.6. At γ = 0.0 we found that the disagreement term contributes lesser in comparison to the recurrent predictor. Interestingly, we obtain a slightly different result at γ = 0.6, RC-GVF with only disagreement (no recurrent predictor) performs better than RC-GVF with only recurrent predictor (no disagreement). This further emphasizes the importance of handling the aleatoric uncertainty with non-zero GVF discounts.
> >
> > The new ablation results are presented with 10 seeds for the complete RC-GVF and 5 seeds for the other variants. We will include final results with 10 seeds in the paper.
> >
> > [Ciosek 2020]  K. Ciosek, V. Fortuin, R. Tomioka, K. Hofmann, and R. E. Turner. Conservative Uncertainty Estimation By Fitting Prior Networks. 8th International Conference on Learning Representations (ICLR) 2020.
> >
> > **“If I understand it correctly, the main difference between RC-GVF with γ = 0 with and without recurrence is that the predictor network unlike in RND is not the same as the pseudo-reward generator and in the case of Minigrid doesn’t use an LSTM core. Did you do an ablation on the extra representational power in the different predictor network architecture of RC-GVF compared to the pseudo-reward generator also for the diabolical lock experiment?”**
> >
> > RC-GVF has a similar predictor (no recurrence) to RND in the diabolical lock experiment.
> > As you mention, RND’s predictor is the same as the pseudo-reward generator. In comparison, the RC-GVF predictor has more capacity (one more hidden layer, and double the neurons in each layer). We had conducted experiments with the high capacity predictor available to RND. This was never successful in solving the lock and suggests that using intrinsic rewards derived from future dynamics is really the key to succeeding in the diabolical lock.
> >
> > We will clarify this in the paper, although we can also run these experiments once more and report quantitative results for them in the appendix if this is desirable.

---

> > > ### Comment · Reviewer_ZFPx · 2022-08-05
> > > **Response to answer of authors**
> > >
> > > I thank the authors for their response and for answering my questions.
> > >
> > > However the ensemble size experiment results are still almost counter-intuitive to me. Why does ensemble size 2 work the best in the KeyCorridorS5R3 experiment? Do the authors have any explanation/intuition why there is no consistent trend across the experiments with varying ensemble sizes? Is this an issue with number of seeds as reviewer 9UDz alluded to?
> > >
> > > Also the authors didn't really comment about my question about the relationship between the noise levels in the environment and the ensemble size. Although the variance between ensemble members should go to zero in noisy states, this on its own is a way of approximating the epistemic uncertainty. And now that I re-read that section, I believe the authors don't discuss this relationship between scaling the prediction error with essentially an epistemic uncertainty estimate well enough.  The authors discuss how the ensemble disagreement vanishes for stochastic states so the prediction error will be multiplied by 0. But what about all the other cases, where ensemble members disagree because of simply high epistemic uncertainty due to lack of data? In that case, you are multiplying the prediction error with an epistemic uncertainty estimate, introducing a new type of relationship that is not discussed. Could that explain why for $\gamma=0$ you see a difference in results for ensemble size 2? Is there a way to check that you are indeed seeing an improvement with an ensemble because of vanishing disagreement at noisy states?
> > >
> > > I also would still like to see quantitative results of a high-capacity predictor for the diabolical lock experiment in an updated version of the supplementary.

---

> > > > ### Author Response · Authors · 2022-08-09
> > > > **Response to Reviewer ZFPx (1/2)**
> > > >
> > > > Thank you for engaging in the discussion period. We reply to your remaining comments and questions below.
> > > >
> > > > **“However the ensemble size experiment results are still almost counter-intuitive to me. Why does ensemble size 2 work the best in the KeyCorridorS5R3 experiment? Do the authors have any explanation/intuition why there is no consistent trend across the experiments with varying ensemble sizes? Is this an issue with number of seeds as reviewer 9UDz alluded to?”**
> > > >
> > > >
> > > > We note that all ensemble sizes (2, 4, 6, 8) perform about the same on KeyCorridorS5R3, and that the natural variance of these algorithms makes it difficult to interpret small differences in performance as are currently observed. Currently, we report 5 seeds for ensemble sizes (4, 6, 8), and 10 for size=2, because of the limited time available during the rebuttal. While this means that the precise curves are still likely to deviate slightly, this is not expected to alter the outcome of this experiment (i.e. that they perform about the same) given the observed variability thus far.
> > > >
> > > > As to why ensembles with fewer members can perform as well as those having additional members, there could be a few reasons:
> > > >
> > > > 1. It might be that because of also having the prediction error term, a precise quantification of the disagreement/variance is not essential.
> > > >
> > > > 2. The ensembles have a common recurrent core and only differ in MLP heads; perhaps separate RNNs for each member would lead to greater variation with ensemble size.
> > > >
> > > > 3. With larger ensemble sizes it may be that additional hyper-parameters need to be tuned separately (at this point, we checked two values of the intrinsic weight coefficient). For instance there could be a dependency through the variance of the sample variance, which could impact the best choices for the intrinsic weight coefficient and the predictor’s learning rate.
> > > >
> > > > **“Also the authors didn't really comment about my question about the relationship between the noise levels in the environment and the ensemble size.”**
> > > >
> > > > Thank you for pointing this out, we apologize for overlooking this question previously and include our response as part of our other reply down below.
> > > >
> > > > **“Although the variance between ensemble members should go to zero in noisy states, this on its own is a way of approximating the epistemic uncertainty."**
> > > >
> > > > We agree that the disagreement term may also approximate epistemic uncertainty on its own. In fact, there are several possibilities for constructing an intrinsic reward within our framework. Our particular choice was motivated by the need to correct a prediction error term, which itself did not separate the aleatoric uncertainty. The error term as a starting point is motivated by the connections to RND, and throughout the development of RC-GVF we profited from being able to study the connections between RND and RC-GVF more closely through this term. How to go from here, and whether the variance term can stand on its own or perhaps should be combined differently with an error term, is an interesting and relevant question for future work. We will update the paper accordingly to discuss this.

---

> > > > > ### Author Response · Authors · 2022-08-09
> > > > > **Response to Reviewer ZFPx (2/2)**
> > > > >
> > > > > **"And now that I re-read that section, I believe the authors don't discuss this relationship between scaling the prediction error with essentially an epistemic uncertainty estimate well enough. The authors discuss how the ensemble disagreement vanishes for stochastic states so the prediction error will be multiplied by 0. But what about all the other cases, where ensemble members disagree because of simply high epistemic uncertainty due to lack of data? In that case, you are multiplying the prediction error with an epistemic uncertainty estimate, introducing a new type of relationship that is not discussed."**
> > > > >
> > > > > Indeed, while the benefit is clear in stochastic states when the prediction error is multiplied by 0, it is more difficult to analyze what will happen in other encounters where multiple different factors interact. In this work, we have set out to investigate this empirically, and our results in ablation that you recommended (figure https://ibb.co/z21MtDD) suggest a clear improvement from using the disagreement term.
> > > > >
> > > > > As mentioned in the reply above, we had originally begun our investigation with the prediction error term only (no ensembled disagreement term). The empirical results we obtained were worse than the current version that adds the disagreement term, which we felt is likely due to incorrectly handling the aleatoric uncertainty when longer horizons are considered. This led us to include the disagreement term which greatly improved our approach.
> > > > >
> > > > > We will update the discussion to include a sketch of possible scenarios where the benefit of the disagreement term is not clear to complement our current discussion, which currently mostly focuses on its benefits in stochastic states. Regarding the analysis suggested in your previous question that we overlooked, we agree that an analysis of different noise levels and ensemble sizes in the Diabolical Lock could be an interesting addition, although the evidence from the ablation that you suggested already provides a strong signal that the inclusion of the variance term is generally useful on representative benchmarks (and which we will include in the paper).
> > > > >
> > > > > **"Could that explain why for γ = 0 you see a difference in results for ensemble size 2?"**
> > > > >
> > > > > A difference between RC-GVF ($\gamma = 0$) and RND in Figure 4c is the inclusion of the multiplicative disagreement term from the ensemble in the former case. This appears to only provide a small benefit (compare to RC-GVF ($\gamma = 0$) w/o disagreement), while history conditioning is the main driver of improvement over RND in the case of ($\gamma = 0$). As to why this small benefit exists, we speculate that simply estimating the intrinsic reward (which includes the prediction error) with multiple predictors may have an advantage in reducing variance.
> > > > >
> > > > > **"Is there a way to check that you are indeed seeing an improvement with an ensemble because of vanishing disagreement at noisy states?”**
> > > > >
> > > > > This is unfortunately difficult to measure, although one piece of evidence we have for this is that _RC-GVF benefits more from the disagreement term than RND alone_ (which we will highlight in the paper upon including these new results).
> > > > >
> > > > > **“I also would still like to see quantitative results of a high-capacity predictor for the diabolical lock experiment in an updated version of the supplementary.”**
> > > > >
> > > > > We re-ran this experiment as requested. The table below presents results of the best RND configuration with the high-capacity predictor on the diabolical lock problem.
> > > > >
> > > > > | Model | Mean and (min, max) of farthest column | Runs that reach end of the lock |
> > > > > | --------- | ---------------------------------------------------- | ----------------------------------------- |
> > > > > | RND     | 32 (11, 60)                              |            0/10                     |
> > > > > | RND (with high-capacity predictor)    |   31 (26, 57)         |            0/10          |
> > > > >
> > > > > It can be seen that even with a high-capacity predictor RND is unable to solve this environment. We will report these results in the supplementary material.

---

### Official Review · Reviewer_MZa2 · 2022-07-12

**Rating:** 6
**Confidence:** 3
**Soundness:** 3 good
**Presentation:** 3 good
**Contribution:** 3 good

**Summary:**

This paper proposes a novel exploration strategy for RL that uses intrinsic rewards based on predicting long term values of random vector values rewards. The proposed approach contains the common RND exploration algorithm as a special case and is shown to work well on standard benchmarks, even when not given privileged access to episodic counts.

**Questions:**

Figure 1: I appreciate the idea of having a simple motivating example, but the details are too sparse. How does the agent get to the end to see the spike? It seems that even predicting future rewards gives no signal until the very end so it would still appear to be a very hard exploration problem without a shaped true reward. What is the statespace here? is it just white and blue? x-pos? What is the task reward?

Line 226: Transitions from "dead" row are unclear. From the current description it seems that  an optimal policy be to take a bad action H-1 times (total reward 0) followed by a good action at the last timestep?

Line 232: I don't understand why you can't just always take the good action. It seems like this deterministic policy works regardless of what state you are in

How sensitive is the algorithm to the weights on true reward and intrinsic reward for RL?


**Limitations:**

yes

**Strengths And Weaknesses:**

+Simple idea that seems to work well and generalizes prior work

+Good results on exploration benchmarks

+Nice use of ensembles to deal with aleatoric and epistemic uncertainty

-The environment details are sparse for someone who hasn't used these benchmarks before

-It is unclear how sensitive the algorithm is to the weights on true reward and intrinsic reward for RL?

---

> ### Author Response · Authors · 2022-08-02
> **Individual response to Reviewer MZa2**
>
> Thank you very much for your review, questions and suggestions for improving the paper. We are glad that you found our idea and results interesting.
>
> We recognize that it can be difficult to understand the considered benchmarks in detail if the reviewer hasn’t used them before. We will add additional details to improve clarity in various parts of the paper based on your feedback (see our response to your questions below). Please let us know in case you encounter any remaining issues in this regard or if any specific details are missing. We provide a point-by-point response to your specific questions below.
>
> **“Figure 1: I appreciate the idea of having a simple motivating example, but the details are too sparse. How does the agent get to the end to see the spike? It seems that even predicting future rewards gives no signal until the very end so it would still appear to be a very hard exploration problem without a shaped true reward. What is the statespace here? is it just white and blue? x-pos? What is the task reward?”**
>
> Thank you for pointing this out. In this motivating example there is no task reward. There is a single action (and only one policy) available to the agent, which is to move forward at every time step. The current state can only be observed through the color of the cell (white or blue), which makes this a partially observable setting. We train our predictors (RND or RC-GVF) online (no buffer) on the data collected in intervals of 2 environment steps during the agent’s interactions.
>
> What we wanted to illustrate with this example is a failure case of RND as it is usually applied in partially observable environments. The RND intrinsic reward yields no ‘surprise’ even though the last sequence of cells is ‘interesting’ in the sense that it breaks away from the repeating pattern of oscillating cell colors. We will clarify these details in the paper.
>
> **“Line 226: Transitions from "dead" row are unclear. From the current description it seems that an optimal policy be to take a bad action H-1 times (total reward 0) followed by a good action at the last timestep?”**
>
> We agree that the current description may erroneously suggest that the optimal policy would be to take a bad action H-1 times. Due to space constraints we were limited to providing further details along with a figure in the supplementary material (Figure 5, in Appendix A.1). We will clarify this with a sentence to communicate that the agent cannot obtain the maximum return once a single bad action is taken.
>
> To avoid any other misunderstandings we will also summarize the dynamics related to the dead row here:
>
> * The transition (for any action) from a dead row in a certain column is to the dead row in the next column with zero reward. Once the agent transitions to the dead row it remains in it until the end of the episode. In the final dead state all actions lead to termination with zero reward.
>
> * It is impossible for the agent to receive a high reward at the end of the lock from a dead state, and hence the optimal policy is to take the respective good actions in each (good) state. The good action at each state is assigned randomly as part of the MDP specification.
>
> **“Line 232: I don't understand why you can't just always take the good action. It seems like this deterministic policy works regardless of what state you are in”**
>
> Indeed, a deterministic policy that selects the good action in every state is the optimal one. However, as we now hope is clear (and what we intended to convey with the sentence in line 232), any fixed/memorized sequence of actions that ignores state can not successfully solve this problem due to the stochastic transitions. The agent must take a correspondingly different action depending on the state it ends up in.
>
> **“How sensitive is the algorithm to the weights on true reward and intrinsic reward for RL?”**
>
> That is an interesting question that we have not explored thus far. This is mainly because we don’t have any reason to suspect that the intrinsic reward generated by RC-GVF requires a different treatment to other intrinsic rewards from the literature in this regard. Similar to all the other baselines we compare with, RC-GVF balances intrinsic and extrinsic rewards through a weight on the intrinsic coefficient. In certain cases like RND, intrinsic rewards would naturally vanish over interactions. This would also be true for NovelD with RND. RC-GVF is closer to recent approaches that propose intrinsic rewards that do not vanish asymptotically [Flet-Berliac et al., 2021; Raileanu et al., 2020]. One way to ensure stability is by annealing the intrinsic coefficient to zero [Flet-Berliac et al., 2021]. In our case, it is simply a small value over which extrinsic rewards will dominate once they are reliably discovered.
>
> We will attempt to include a basic investigation into this matter in the form of an experiment with different mixing weights.

---

> > ### Comment · Reviewer_MZa2 · 2022-08-08
> > **response**
> >
> > Thank you for your clarification and answers to my questions. I agree that doing a sensitivity analysis on the weighting of true and intrinsic reward would be valuable. Also, making it clear how they are weighted and/or annealed in the main paper would be good.

---

### Author Response · Authors · 2022-08-02
**General response**

We would like to thank the reviewers for their thoughtful comments and positive feedback. We are pleased that the reviewers found our approach novel or interesting (Reviewers MZa2, ZFPx, 5dYH), well-motivated (Reviewers 9UDz, 5dYH), and simple, intuitive or fundamentally well justified (Reviewers MZa2, 9UDz, 5dYH).

In response to the reviewers feedback we have conducted several additional experiments and analysis, which we will incorporate in the paper. We summarize the main experiments and findings below.

**Ensemble size ablations (Reviewers ZFPx and 5dYH).**

We conducted experiments with larger ensemble sizes on the MiniGrid and Diabolical Lock experiments. We observed a small improvement with increased ensemble size in diabolical lock, where we present results of lock completion at 15M frames with 2 (original), 4 and 6 members in the ensemble as in Table 1:

| # predictors | Mean and (min, max) of farthest column | Runs that reach end of the lock |
| --------------- | ---------------------------------------------------- | ----------------------------------------- |
|         2          |   		94 (66, 100)		      |			6/10		|
|          4	           | 			98 (90,100)		     |			8/10		|
|          6 	|		95 (65, 100)		    |  			8/10		|

In two MiniGrid environments, we obtained comparable results with all ensemble sizes (KeyCorridor-S5R3: https://ibb.co/X5bKCYb and MultiRoom-N12S10: https://ibb.co/L8qbDH9). These new findings indicate that two member ensembles usually suffice in these problems.

**Additional baselines on Diabolical Lock (Reviewer 5dYH).**

 We added a comparison of NovelD and AGAC to RC-GVF on the Diabolical Lock. We present the results for lock completion at 15M frames for these approaches and RC-GVF in the table below.

| Model | Mean and (min, max) of farthest column | Runs that reach end of the lock |
| --------------- | ---------------------------------------------------- | ----------------------------------------- |
| RC-GVF      |   		94 (66, 100)		      |			6/10		|
|          ACAG           | 		5 (4, 6)		     |			0/5	|
|          NovelD 	|		24 (22, 31)		    |  			0/5		|

It can be observed how AGAC and NovelD perform poorly on this challenging exploration task, further emphasizing the need for longer horizon prediction as in RC-GVF.

**Additional ablations of ensembling and history conditioning (Reviewer ZFPX).**

We conducted additional ablations of RC-GVF at γ = 0.6 on the KeyCorridor-S5R3 environment. Concretely, we removed either the recurrent predictor or the disagreement term. Our results (figure: https://ibb.co/z21MtDD) indicate that both factors are important at γ = 0.6, and removing either of them degrades performance. RC-GVF with only disagreement (no recurrent predictor) performs better than RC-GVF with only recurrent predictor (no disagreement). We provide additional insight into these new findings in the response to reviewer ZFPx.

**RC-GVF with episodic state counts (Reviewer 9UDz).**

We now report results for RC-GVF using ground-truth episodic state counts on KeyCorridor-S5R3 (https://ibb.co/Gx0PJrN) and  MultiRoom-N12-S10 (https://ibb.co/VCrcCcv). While RC-GVF achieves competitive performance in the former environment (as expected), we observe interference between the GVF prediction and the episode state counts in the latter environment. Further discussion is in the response to reviewer 9UDz.

**Analysis of discount factors (Reviewer 9UDz).**

We expand our analysis of RC-GVF to include larger GVF discount factors (0.99 and 0.95). We observed that 0.95 works well in the KeyCorridor-S5R3 environment (figure https://ibb.co/SmWsNLH), but not in MultiRoom-N12-S10 (figure https://ibb.co/bHBQDXR). 0.99 was worse than our default discount factor of 0.6 on both environments. These results indicate that while longer horizon prediction is important, extending the horizon too much may make the prediction problem too difficult. Further discussion is in the response to reviewer 9UDz.

We respond to individual points raised by reviewers in separate replies to each review below.

---

### Meta-Review · Area_Chair_bK6N · 2022-08-26

**Recommendation:** Accept
**Confidence:** Certain

**Metareview:**

This paper proposes an intrinsic motivation method which by and large extends RND, aiming to service the needs of agents in longer horizon exploration problems. There was a bit of a spread of scores amongst reviewers, but based upon reading the extensive discussion, I am recommending acceptance it seems that on the balance of opinions, the consensus leans towards the reviewers agreeing that the result is important, the method useful, and the paper is of a publishable nature.

**Award:**

No

---

### Decision · Program_Chairs · 2022-09-14

Accept